



# Quantifying cascading uncertainty in compound flood modeling with linked process-based and machine learning models

David F. Muñoz[1,2,3], Hamed Moftakhari[3,4], Hamid Moradkhani[3,4]

[1]Department of Civil and Environmental Engineering, Virginia Tech, Blacksburg, VA 24060, United
States
[2]Center for Coastal Studies, Virginia Tech, Blacksburg, VA 24060, United States
[3]Center for Complex Hydrosystems Research, The University of Alabama, Tuscaloosa, AL 35487,
United States
[4]Department of Civil, Construction and Environmental Engineering, The University of Alabama,
Tuscaloosa, AL 35487, United States

*Correspondence to*: David F. Muñoz (davidmunozpauta@vt.edu)

**Abstract.**

Compound flood (CF) modeling enables the simulation of nonlinear water level dynamics in which
concurrent or successive flood drivers synergize, producing larger impacts than those from individual

drivers. CF modeling is yet subject to four main sources of uncertainty including (i) initial condition,
(ii) forcing (or boundary) conditions, (iii) model parameters, and (iv) model structure. These sources
of uncertainty, if not quantified and effectively reduced, cascade in series throughout the modeling
chain and compromise the accuracy of CF hazard assessments. Here, we characterize cascading
uncertainty using linked process-based and machine learning (PB-ML) models for a well-known CF

event, namely Hurricane Harvey in Galveston Bay, TX. For this, we run a set of hydrodynamic model
scenarios to quantify isolated and cascading uncertainty in terms of maximum water level residuals,
and additionally, track the evolution of residuals during the onset, peak, and dissipation of Hurricane
Harvey. We then develop multiple-linear regression (MLR) and PB-ML models to estimate the relative
and cumulative contribution of the four sources of uncertainty to total uncertainty over time. Results

from this study show that the proposed PB-ML model capture "hidden" nonlinear associations and
interactions among the sources of uncertainty, thereby outperforming conventional MLR models.



Model structure and forcing conditions are the main sources of uncertainty in CF modeling and their corresponding model scenarios, or input features, contribute to (56%) 49% of variance reduction in the estimation of (maximum) water level residuals. Following these results, we conclude that PB-ML models are a feasible alternative for quantifying cascading uncertainty in CF modeling.

## 1 Introduction

It is estimated that nearly half of the gross domestic product in the U.S., i.e., 46% of GPD, is generated in coastal shoreline counties that are frequently exposed to multiple flood hazards (NOAA Digital Coast, 2020). Similarly, nearly 129 million people in the U.S. (39% of its population) currently live in low-lying areas at risk of inland and coastal flooding (NOAA, 2022). In the past five years, the National Center for Environmental Information reported 489 fatalities and over $327 billion of total cost damages as a result of tropical cyclones; in which heavy rainfall and storm surge exacerbate coastal flood impacts (NCEI, 2023). Terrestrial and coastal drivers of (non-) extreme nature that either coincide or unfold in close succession trigger compound flood (CF) events such as those already evinced in the U.S. history, i.e., Hurricane Katrina (2005), Sandy (2012), Harvey (2017), Florence (2018), Ida (2021), Ian (2022), and Idalia (2023). CF events in low-lying areas are typically associated with tropical cyclones for which rainfall-runoff, wind-driven storm surge, or both can be classified as dominant flood hazard drivers (Bevacqua et al., 2020; Eilander et al., 2020). In addition, the role of waves, tides, and nonlinear interactions on extreme water levels (WLs) can be crucial for the accurate simulation and/or prediction of CF events as reported in several studies (Hsu et al., 2023; Nasr et al., 2021; Serafin et al., 2017).

CF modeling can be performed via multivariate statistical analysis (Bensi et al., 2020; Jalili Pirani and Najafi, 2023; Sadegh et al., 2018), process-based modeling (Bates et al., 2021; Sanders et al., 2023; Santiago-Collazo et al., 2019), and even "hybrid" methods (Gori et al., 2020; Moftakhari et al., 2019; Serafin et al., 2019). Statistical analyses enable the prediction of future CF events, the





reliability of which largely depends on the length of data records. This means that a detailed CF hazard assessment over a given spatial domain requires the availability of both data records and computational resources for handling large datasets (Eilander et al., 2023). For hindcasting purposes, CF events are simulated using process-based models as they can incorporate physical features in the underlying digital elevation model (DEM) including local hydrodynamic attributes and geomorphologic characteristics, i.e., tidal and riverine channels, artificial waterways, and flood infrastructure (Marsooli and Wang, 2020; Muñoz et al., 2020; Salehi, 2018). Another advantage of process-based modeling is the ability to simulate complex WL dynamics such as backwater effects, tidal propagation, and overtopping in estuarine environments and urban settings that are usually ignored in point-based statistical analyses (Gallien et al., 2018; Kumbier et al., 2018; Leijnse et al., 2021). Also, process-based models can simulate complex CF dynamics in coastal to inland transition zones where hydrological and coastal processes determine flood extent, duration, and inundation depth (Bilskie et al., 2021; Jafarzadegan et al., 2023; Peña et al., 2022). Nevertheless, CF modeling is subject to uncertainties that interact and cascade in series throughout the modeling chain if they are not treated appropriately (Beven et al., 2005; Hasan Tanim and Goharian, 2021; Meresa et al., 2021).

In general, uncertainties in process-based modeling can be classified into four main sources: (i) initial condition, (ii) forcing (or boundary) conditions, (iii) model parameters, and (iv) model structure (Beven et al., 2005; Moradkhani et al., 2018; Vrugt, 2016). Initial and forcing conditions are essentially model inputs to any process-based models, however, their isolated effects on WL dynamics are often analyzed separately as reported in diverse hydrological (Abbaszadeh et al., 2019; Jafarzadegan et al., 2021a; Kohanpur et al., 2023) and coastal studies (Bakhtyar et al., 2020; Marsooli and Wang, 2020; Muñoz et al., 2022a). The first source of uncertainty involves inaccuracies in the geometry of the system, which is spatially represented with light detection and ranging (LiDAR) elevation data. These inaccuracies also include bathymetric (Cea and French, 2012; Neal et al., 2021; Parodi et al., 2020) and topographic errors such as those reported in tidal wetland regions (Alizad et al., 2018; Cooper et al., 2019; Rogers et al., 2018). Elevation errors in coastal wetlands can reach



values up to 0.65 m and are usually estimated as the vertical difference between LiDAR-derived DEMs and ground-truth elevation collected during real-time kinematic surveys (Medeiros et al., 2015; Rogers et al., 2016). Uncertainty stemming from forcing or boundary conditions is linked with instrument and

sensor's characteristics that measure WL or streamflow such as analog-to-digital recorders and acoustic doppler current, respectively (NOAA, 2000; USGS, 2011). Notably, this uncertainty can also arise from *a posteriori* assumption (or generalizations) in operational hurricane-induced coastal flood forecasting. For example, the Coastal Emergency Risk Assessment portal provides real-time storm-surge, wave, and flood guidance for the Gulf and Atlantic Coasts of the U.S. under the assumption that

river flow and local rainfall contributions to flooding are relatively small as compared to that driven by storm surge (CERA, 2023). Although this assumption might be valid for non-estuarine regions, ignoring nonlinear interactions among flood drivers in freshwater-influenced stretches of the coast can lead to an underestimation of CF hazards especially in coastal to inland transition zones characterized by tidally-influenced rivers (Bakhtyar et al., 2020; Muñoz et al., 2022b; Yin et al., 2021).

Another important source of uncertainty in CF modeling is associated with model parameters such as the Manning's roughness coefficient that is present in the bottom stress component of the momentum equation (see Section 2.3). Manning's roughness help account for bed friction exerted by vegetation, seabed, riverbed, sinuosity, and irregularity of channel cross-sections (Attari and Hosseini, 2019; Bhola et al., 2019; Yen, 2002). Thus, hydrodynamic models rely on a rigorous static (or

dynamic) calibration of roughness coefficients to capture the onset, peak, and dissipation of WLs as well as CF dynamics (Jafarzadegan et al., 2021a; Liu et al., 2018; Mayo et al., 2014). However, conducting model calibration is a computationally intensive procedure that requires a suitable strategy to explore and exploit the parameter space such as Monte Carlo and Latin Hypercube Sampling techniques (Helton and Davis, 2003; Kuczera and Parent, 1998). For that reason, flood hazard

assessments often assume stationarity of model parameters under the premise that calibrated roughness coefficients for a specific event are adequate for a range of unseen flood scenarios (Domeneghetti et al., 2013; Meresa et al., 2021). The fourth source of uncertainty refers to limitations or *a priori*



(theoretical) assumptions that are necessary to simplify the representation of oceanic, hydrological, and meteorological processes in regard to flood generation and routing (Moradkhani et al., 2018; Nearing et al., 2016; Pappenberger et al., 2006). Moreover, uncertainty derived from model structure accounts for model configuration and so inherent "reduced-physics" schemes to solve the conservation of mass and momentum equations (see Section 2.3). For example, reduced-physics numerical schemes are devised to ignore local acceleration, pressure gradient, viscosity, and/or Coriolis terms (Brunner, 2016; Leijnse et al., 2021; Lesser et al., 2004). Nonetheless, such schemes are designed to optimize the modeling procedure, i.e., reduce the computational cost or time required by high-fidelity process-based models while ensuring an acceptable accuracy in the simulation of WL and CF dynamics.

Methods for uncertainty quantification vary in complexity and application and have been discussed in detail in recent review studies (Abbaszadeh et al., 2022; Beven et al., 2018; Xu et al., 2023). Those methods include linear associations and first-order second moment approximations (Taylor et al., 2015; Thompson et al., 2008), generalized likelihood estimations (Aronica et al., 2002; Domeneghetti et al., 2013), sensitivity analyses (Alipour et al., 2022; Hall et al., 2005; Savage et al., 2016), multi-model ensemble methods (Duan et al., 2007; Madadgar and Moradkhani, 2014; Najafi and Moradkhani, 2016), and data assimilation (Abbaszadeh et al., 2019; Moradkhani et al., 2018; Pathiraja et al., 2018).

In recent years, researchers have explored linked process-based and machine learning (PB-ML) models for uncertainty analysis. Hu et al., (2019) developed an integrated framework consisting of ML and reduced order models for rapid flood prediction and uncertainty quantification. Specifically, they reported that forcing conditions (e.g., incoming waves) are the main source of uncertainty for predicting water surface elevation resulting from tsunamis. Moreover, they quantified such an uncertainty via prescriptive analytics in long short-term memory (LSTM) networks, i.e., inverse functions. Anaraki et al., (2021) proposed a hybrid modeling framework that combines hydrological models and ML for flood frequency analysis under climate change conditions. They indicated that the selection of hydrological models (e.g., model structure) is a critical source of uncertainty based on





fuzzy and analysis of variance methods. Chaudhary et al., (2022) developed a deep learning ensemble

model that is trained with hydrodynamic model outputs to predict urban flood hazards at high spatial

resolution. They estimated total predictive uncertainty in terms of aleatory and epistemic uncertainty

by focusing on model inputs and model parameters (e.g., deep learning model's weights). Also, they

reported that both sources of uncertainty follow the pattern of maximum water depth residuals and that

aleatory and epistemic uncertainty are sharper and fuzzier for higher residual values, respectively.

## 135    2 Materials and methods

### 2.1 Study area

We select Galveston Bay (G-Bay) as the study area to leverage multiple spatiotemporal datasets

and official reports that help calibrate and validate hydrodynamic models (East et al., 2008; Rego and

Li, 2010; Sebastian et al., 2021; Valle-Levinson et al., 2020). G-Bay is the seventh largest estuary in

the U.S. that connects Houston, TX with the Gulf of Mexico trough a complex system consisting of

bayous, interior bays, and rivers (Figure 1a). G-Bay is a shallow estuary of 2 m depth, 56 km length,

and 31 km width (on average) that comprises an area of 1600 km$^2$, approximately. Annual average

freshwater flow into the G-Bay comes from the Buffalo Bayou River and their tributaries (USGS

08074000) and the San Jacinto River (Lake Houston's dam) is 50 m$^3$/s and 75 m$^3$/s, respectively

(Figure 1b and 1c). Tides reaching the G-Bay entrance (NOAA 8771341) are mixed and characterized

by the lunar diurnal (K$_1$) and principal lunar semidiurnal (M$_2$) constituents with tidal amplitudes of

0.15 m and 0.11 m, respectively.

We simulate two CF events in G-Bay, namely Hurricane Ike and Hurricane Harvey, that hit

the Gulf of Mexico in September 2008 and August 2017, respectively (Figure 1a). Hurricane Ike made

landfall as Category 2 in the Saffir-Simpson scale in the eastern part of Galveston Island, TX on

September 13, 2008. Ike produced storm surges up to 4 m near Sabine Pass and 480 mm of cumulative

precipitation over southeastern TX that together led to maximum inundation depths up to 3 m above

ground level in Galveston County (Berg, 2009; Rego and Li, 2010). Hurricane Harvey, on the other
hand, reached Category 4 near Rockport, TX on August 24, 2017 and made a second landfall near

Cameron, LA on August 29, 2017. Harvey generated total cumulative precipitations ranging from 0.64
m up to 1.52 m over southeastern Texas and subsequent pluvial flooding in the upper river reaches of
the Buffalo Bayou with maximum inundation depths of 3 m above ground level (Blake and Zelinsky,
2018). In addition to heavy rainfall, wind-driven storm surge triggered compound coastal flooding
over the region that lasted 3 to 8 days (Huang et al., 2021; Valle-Levinson et al., 2020).


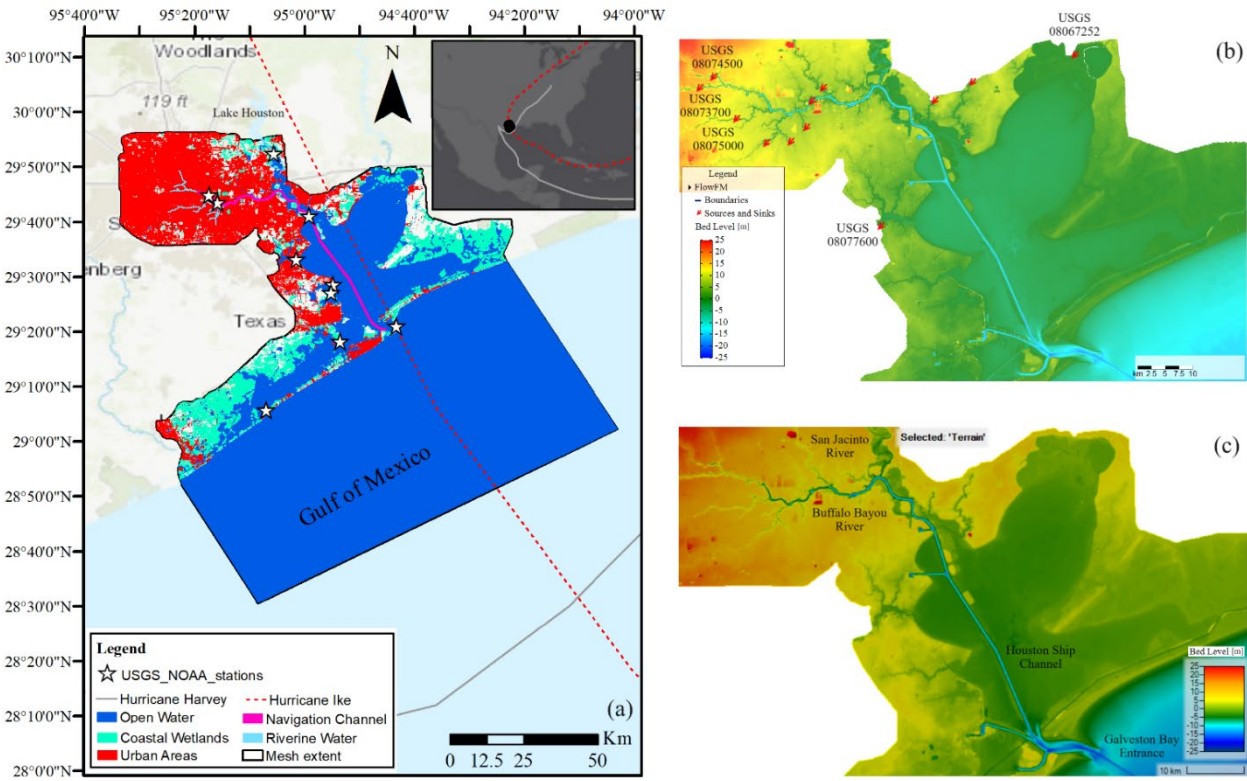

**Figure 1.** Model domain of Galveston Bay, TX. (a) Tide-gauge stations and land cover categories
derived from the National Land Cover Database. Solid and dashed lines illustrate the best tracks of



Hurricane Ike (Sep, 2009) and Harvey (Aug, 2017), respectively. Topography and bathymetry of the
study area interpolated in the (b) Delft3D-FM and (c) 2D HEC-RAS grid.

## 2.2 Data availability

We use publicly available data to develop and calibrate hydrodynamics models of G-Bay
(Figure 1b and 1c). To resemble physical conditions prior to Hurricane Ike, we consider the legacy
Galveston Texas Coastal DEM obtained from the NOAA's National Geophysical Data Center
(https://www.ncei.noaa.gov/access/metadata/landing-page/bin/iso?id=gov.noaa.ngdc.mgg.dem:403).
This DEM includes topographic and bathymetric (topobathy) data of 2006 and is referenced to the
vertical tidal datum of Mean High Water (MHW). Next, we resemble physical conditions prior to
Hurricane Harvey by considering a legacy continuously updated digital elevation model (CUDEM)
that includes topobathy data of 2017. The CUDEM is referenced to the North American Vertical
Datum 1988 (NAVD88) and can be obtained from the NOAA's Data Access Viewer
(https://coast.noaa.gov/). To account for wetland elevation errors in the CUDEM, we consider a
recently published coastal DEM that provides relative tidal marsh elevations over the Conterminous
United States at 30 m spatial resolution and referenced to the MHW datum (Holmquist and Windham-
Myers, 2022). It is important to note that the coastal DEMs are referenced to the NAVD88 datum using
the NOAA's Vertical Datum tool (https://vdatum.noaa.gov/). Likewise, we consider land cover maps
derived from the 2008 and 2016 National Land Cover Database (NLCD) as a proxy of spatially
distributed roughness values for model calibration of Hurricane Ike and Harvey, respectively
(https://www.mrlc.gov/data). The NLCD maps have a 30 m spatial resolution and 16 land cover classes
over the continental U.S. that can be conveniently regrouped or aggregated into general categories to
avoid unnecessary specificity during model calibration.

Forcing or forcing conditions (BCs) consist of time-series data of water level (WL) and river
discharge    that    are    obtained    from    the    NOAA's    Tide    &    Currents    portal
(https://tidesandcurrents.noaa.gov/)    and    the    USGS'    National    Water    Dashboard



(https://dashboard.waterdata.usgs.gov/), respectively. In addition, we force the model at the ocean
boundary using barotropic tides obtained from the TPXO 8.0 Global Inverse Tide Model
(https://www.tpxo.net/global/tpxo8-atlas). To evaluate hydrodynamic model's performance, we
leverage survey data from a temporary monitoring network deployed for Hurricane Ike, i.e., water
pressure sensors (East et al., 2008), and post-flood high-water marks from the USGS's Flood Event
Viewer (https://stn.wim.usgs.gov/fev/) (Figure 1a). Rainfall data are obtained from the rain-gauge
network of the Harris County - Flood Warning System (https://www.harriscountyfws.org/). In
addition, we use gridded data from ERA5 reanalysis dataset to account for hourly local wind,
atmospheric     pressure,     and     total     precipitation     at     30     km     spatial     resolution
(https://www.ecmwf.int/en/forecasts/datasets/reanalysis-datasets/era5).

## 2.3 Hydrodynamic modeling

We develop hydrodynamic models using two different model software in order to simulate
compound coastal flooding. We then analyze the uncertainty stemming from model structural
inadequacy reflected in model configuration and numerical scheme. Specifically, we setup two-
dimensional (2D) models in Delft3D-Flexible Mesh (FM) version 2021.3 and the U.S. Army Corps of
Engineers' River Analysis System (HEC-RAS) version 6.3. Both models are widely used in pluvial,
fluvial, and coastal flood studies and have achieved satisfactory results (Bakhtyar et al., 2020; Liu et
al., 2018; Muñoz et al., 2021; Shustikova et al., 2019). Delft3D-FM can be setup in 2D (depth-
averaged) mode to solve the continuity (Eq. (1)) and Reynolds-averaged Navier-Stokes equations (Eq.
(2) and (3)) for incompressible fluids, uniform density, and vertical length scales that are significant
smaller that the horizontal ones (Lesser et al., 2004; Roelvink and Van Banning, 1995). In a similar
way, HEC-RAS solves 2D unsteady flow and recent model developments (e.g., version 6.3 onwards)
include gridded wind and precipitation forcing input in the momentum conservation equations
(USACE, 2023).



$$\frac{\partial \zeta}{\partial t} + \frac{\partial (d + \zeta)u}{\partial x} + \frac{\partial (d + \zeta)v}{\partial y} = 0 \tag{1}$$

$$\frac{\partial u}{\partial t} + u\frac{\partial u}{\partial x} + v\frac{\partial u}{\partial y} + fv$$
$$= \frac{\rho_a C_d (U_{10} - u)}{\rho(d + \zeta)} [(U_{10} - u)^2 + (V_{10} - v)^2]^{\frac{1}{2}} - \frac{\rho g n^2}{d^{\frac{1}{3}}} u(u^2 + v^2)^{\frac{1}{2}} + v_H \left| \frac{\partial^2 u}{\partial x^2} + \frac{\partial^2 u}{\partial y^2} \right| - g\frac{\partial \zeta}{\partial x} \tag{2}$$

$$\frac{\partial v}{\partial t} + u\frac{\partial v}{\partial x} + v\frac{\partial v}{\partial y} - fu$$
$$= \frac{\rho_a C_d (V_{10} - v)}{\rho(d + \zeta)} [(U_{10} - u)^2 + (V_{10} - v)^2]^{\frac{1}{2}} - \frac{\rho g n^2}{d^{\frac{1}{3}}} v(u^2 + v^2)^{\frac{1}{2}} + v_H \left| \frac{\partial^2 v}{\partial x^2} + \frac{\partial^2 v}{\partial y^2} \right| - g\frac{\partial \zeta}{\partial y} \tag{3}$$

where $\zeta$ is water surface elevation (above still water), $t$ is time, $d$ is water depth (below horizontal datum or still water), $u$ and $v$ are 2D depth-averaged velocities in x and y directions, $f$ is the Coriolis parameter, $\rho_a$ is air density, $C_d$ is the wind-drag coefficient, $\rho$ is water density, $U_{10}$ and $V_{10}$ are wind velocities at 10 m height above still water in x and y directions, $g$ is the gravitational acceleration, $n$ is the Manning's roughness coefficient, and $v_H$ is the horizontal viscosity parameter. Note that unlike Delft3D-FM, HEC-RAS does not account for the atmospheric pressure term in Eq. (1) and (2).

### 2.3.1 Model setup

The first hydrodynamic model is developed in Delft3D-FM using an unstructured finite volume grid that consists of triangular cells with spatially varying size. Unstructured grids help capture geomorphological and urban features with greater detail than conventional nested structured grids (Kumar et al., 2009; Muñoz et al., 2022a). Those features include the Galveston Bay entrance, artificial channels in Houston, intracoastal waterways, lateral floodplains, wetland regions, and bottleneck-like connections between G-Bay and both the Buffalo Bayou River and San Jacinto River (Figure 1c). Triangular cell sizes are set to increase from 3 km at the open ocean boundary in the Gulf of Mexico up to 5 m in Harris County. This ensures that CF dynamics are simulated with sufficient detail in both natural and urban environments. Similarly, the second hydrodynamic model is developed in in 2D HEC-RAS using an unstructured finite volume grid. The mesh consists of polygons of varying cell



size and the numerical scheme to solve the shallow water equations is set to the Eulerian-Lagrangian (SWL-ELM) method. This in turn ensures the model solves for all terms in Eq. 1 to 3, except for atmospheric pressure due to the current model capabilities of 2D HEC-RAS. In addition, we force the mesh generation with a cell size and spatial distribution similar to that of Delft3D-FM model. Although there is no a straightforward procedure to transfer the mesh properties and/or spatial characteristics

between the two hydrodynamic models, we ensure that geomorphological and urban features are correctly delineated by conducting an extensive mesh refinement in critical locations as suggested in similar studies (Muñoz et al., 2021; Shustikova et al., 2019). The time step is controlled by the Courant Friedrichs-Lewy condition with a maximum value of 0.7 for both models. Also, model outputs are generated with an hourly interval for calibration and validation purposes.

After the mesh generation process, we consider multiple USGS's river discharge stations in the G-Bay model as upstream BCs including Whiteoak Bayou (USGS 08074500), Buffalo Bayou (USGS 08073700), Brays Bayou (USGS 08075000), Sims Bayou (USGS 08075500), Berry Bayou (USGS 08075605), Greens Bayou (USGS 08076000), Hunting Bayou (USGS 08075770), Vince Bayou (USGS 08075730), Clear Creek (USGS 08077600), Goose Creek (USGS 08067525), Cedar Creek

(USGS 08067500), and the Trinity River (USGS 08067252). Also, river flow from the Lake Houston Dam to the San Jacinto River is estimated as the sum of upstream freshwater input to the lake due to the lack of available river-gauge stations located immediately downstream the dam (Figure 1). Such an estimation is realistic since the dam is not operating as a flood control structure anymore and it was overflowed by extreme river discharge as a result of Hurricane Harvey (Sebastian et al., 2021; Valle-

Levinson et al., 2020). Regarding the downstream BCs, we force the G-Bay model with storm tides using WL records from two tide-gauge stations, namely Freeport Harbor (NOAA 8772471) and Galveston Bay Entrance (NOAA 8771341). These stations are located offshore and are a good proxy for coastal WL propagating from the open ocean boundary. To account for WL variability arising from atmospheric variables, we include reanalysis data of 10 m wind velocity and atmospheric (sea level)

pressure in the model simulations. Note that the latest versions of 2D HEC-RAS allow the user to



simulate wind speeds even though the atmospheric pressure component is yet to be implemented in Eq. 2 and 3.

Lastly, we retrieve rainfall data from a relative dense rain-gauge network of the Harris County – Flood Warning System. These data have been validated and further used to estimate flood damages in G-Bay (Sebastian et al., 2021). In addition, we complement these data with "total precipitation" reanalysis datasets from ERA-5 in order to estimate rainfall patterns in coastal areas beyond the Harris County and over the Gulf of Mexico (Figure 1). Specifically, we use the "inverse distance weight" as the interpolation method in ArcGIS with an output cell size of 1 km (e.g., shortest Euclidean distance between existing rain gauges), search radius of 5 points, and a power function of 2. The interpolation method as well as the aforementioned values follow those suggested in other studies and are validated through sensitivity analysis (Ahrens, 2006; Otieno et al., 2014; Sebastian et al., 2021).

### 2.3.2 Model calibration and validation

G-Bay is influenced by multiple flood drivers including local rainfall, river discharge, and storm tides. At low river flow rates, tides propagate from the ocean boundary in landward direction where they attenuate and eventually vanish due to bottom friction (Bolla Pittaluga et al., 2015; Hoitink and Jay, 2016). We therefore ensure that the model setup (Section 2.3.1) is adequate to simulate tidal propagation across the model domain. Specifically, we simulate tidal dynamics in Delft3D-FM by setting barotropic tides from the TPXO 8.0 Global Inverse Tide Model as forcing data in the open ocean boundary (Egbert and Erofeeva, 2002). Then, we run 100 ensemble model realizations for a 1-year simulation window without incorporating any additional forcing (e.g., only tides) and considering a range of plausible Manning's roughness values ($n$) derived from pertinent literature and technical reports (Arcement and Schneider, 1989; Liu et al., 2018). Combination of plausible $n$-values are generated with the Latin Hypercube Sampling (LHS) technique (Helton and Davis, 2003) and evaluated via ensemble model simulations in a high-performance computing (HPC) system (Table 1). Then, we identify an optimal (calibrated) value for the "open water" category that achieves the lowest





Root-Mean Square Error (RMSE) and Mean Absolute Error (MAE) as well as the highest Kling-Gupta
Efficiency (KGE) and Nash-Sutcliffe Efficiency (NSE).

**Table 1.** Manning's roughness values tested for model calibration in Delft3D-FM / 2D HEC-RAS.

| Land cover category | Open water | Navigation channel | Riverine water | Coastal wetlands | Urban areas |
|---|---|---|---|---|---|
| Lower limit | 0.005 | 0.005 | 0.01 | 0.025 | 0.020 |
| Upper limit | 0.035 | 0.150 | 0.150 | 0.150 | 0.070 |
| Optimal (only tides) | | - | - | - | - |
| Optimal (Ike) | 0.015 | 0.019/0.017 | 0.019/0.018 | 0.029/0.019 | 0.032/0.041 |
| Optimal (Harvey) | | 0.011/0.015 | 0.037/0.019 | 0.051/0.030 | 0.030/0.049 |


Furthermore, we conduct harmonic analyses and retrieve tidal amplitudes and phases for the
main tidal constituents in G-Bay including K2, S2, M2, N2, K1, S1, P1, O1, and Q1 (NOAA, 2000).
Then, we compare observed and simulated tidal characteristics at each tide-gauge station using a 1:1
line assessment as well as concentric circles. In general, the evaluation metrics suggest that the model
setup is adequate to simulate tidal propagation in G-Bay. Also, the optimal *n*-value for open water
helps minimize both RMSE and MAE and achieve high NSE and KGE scores (Figure 2a). The 1:1
line shows a satisfactory agreement between observed and simulated tidal amplitudes with a maximum
MAE of 1.53 cm (Figure 2b). Likewise, observed and simulated tidal phases are in good agreement
for the majority of tidal constituents and stations in G-Bay. The closer the colored (simulated) and gray
(observed) markers in each concentric circle, the more accurate the simulated phase. Note that we only
evaluate tidal propagation using Delft3D-FM model as this software is installed in our HPC system
whereas 2D HEC-RAS is run in a desktop computer. Nevertheless, we expect similar score metrics
regarding tidal propagation for a 1-year simulation window since the model setup of 2D HEC-RAS is
similar to that of Delft3D-FM, i.e., mesh extent, refinement regions, and cell size grid. Also, we support





this claim based on calibration and validation results of CF events as explained later on, i.e., Hurricane

Ike and Harvey.

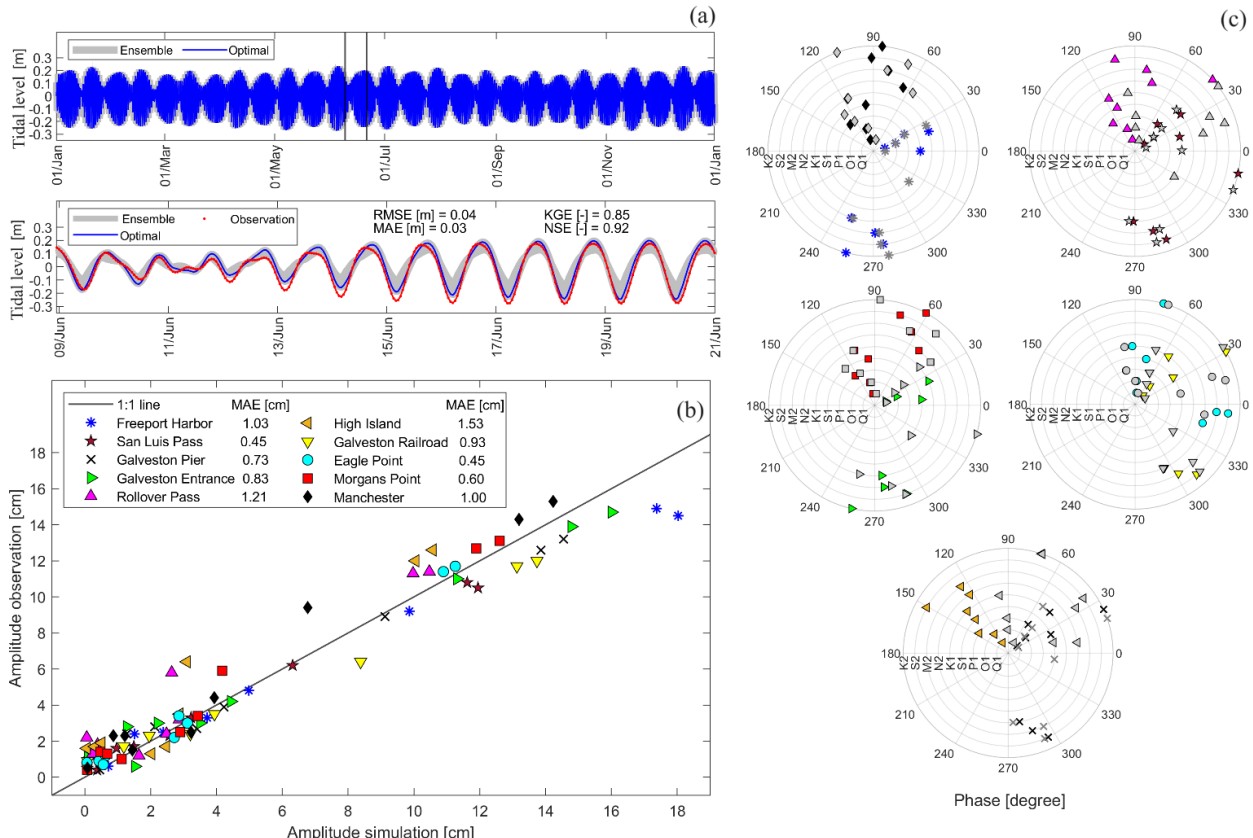

**Figure 2.** Evaluation of tidal propagation in Galveston Bay. (a) Ensemble and optimal model simulation of tides for a 1-year window and a zoom-in section (vertical lines) showing the observed
tidal levels. Results of harmonic analysis show (b) tidal amplitudes (1:1 line), and (c) tidal phase for selected tidal constituents (concentric circles). Each colored marker represents a tide-gauge station with the underlying tidal constituents whereas gray markers represent observed tidal phases.

     Next, we calibrate $n$-values of the navigation channel, riverine water, coastal wetlands, and
urban areas for Hurricane Ike and Harvey (Table 1). We follow an identical process to identify optimal



*n*-values for both Delft3D-FM and 2D HEC-RAS models, i.e., LHS technique with 100 ensemble members, but this time we keep invariant the calibrated *n*-value for the "open water" category to ensure an accurate tidal propagation. It is worth noting that both models are independently calibrated in order to ensure a reliable assessment of model parameters and model structure errors, especially in riverine areas in Houston. The roughness coefficient of open water is common for both models as it simulates tidal propagation from the ocean boundary. Also, we set a 1-month warm-up period for the model to reach equilibrium and consider a 1-week simulation window (centered on the peak WL) to assess the accuracy of model simulations (Figure 3 and Supplementary figures S1 – S2). In general, both hydrodynamic models perform satisfactorily especially at downstream tide-gauge stations where RMSE is below 0.30 cm. However, RMSE progressively increases at upstream tide-gauge stations due to a relatively large riverine influence and underlying nonlinear interactions with storm tides (Figure 3d and 3h). The latter is more evident in model simulations of Hurricane Harvey as the peak river discharge in the Buffalo Bayou station (USGS 08073700), located upstream the Manchester tide-station (NOAA 8770777), is 3.5 times higher than that of Hurricane Ike. Also, inaccuracies of topobathy data along the Buffalo Bayou and river tributaries might have reduced model's performance in the upstream part of G-Bay as reported in flood hazard and damage studies (Jafarzadegan et al., 2021a; Valle-Levinson et al., 2020; Wing et al., 2019).

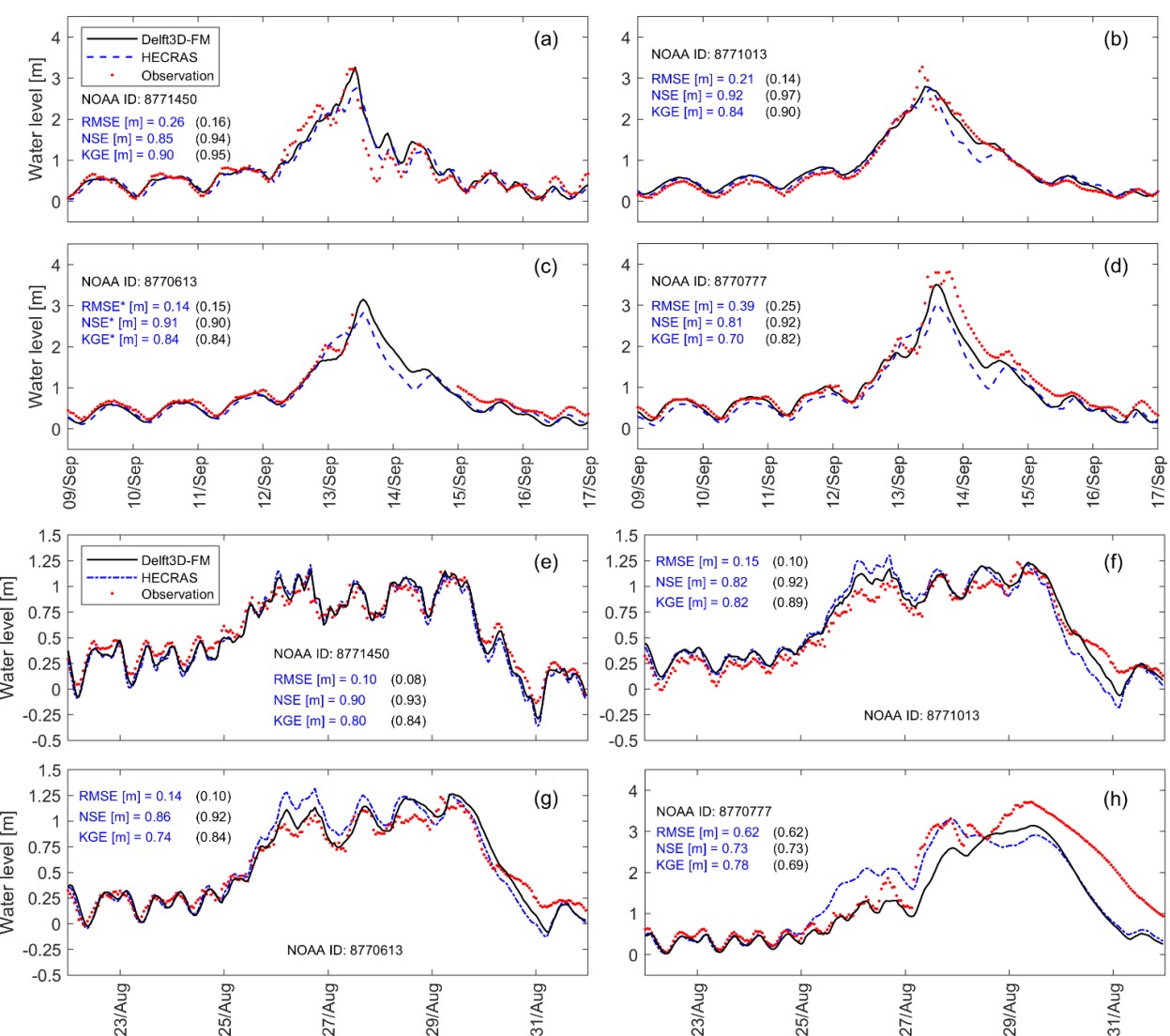

**Figure 3.** Model calibration at selected tide-gauge stations in Galveston Bay. Model performance is evaluated in terms of RMSE, NSE, and KGE for (a-d) Hurricane Ike, and (e-h) Hurricane Harvey. Color code indicate score metrics for Delft3D-FM (black) and 2D HEC-RAS (blue).

To validate the hydrodynamic models, we generate composite maps representing maximum WLs within the simulation period of Hurricane Ike and Harvey and compare those maps against





USGS' high-water marks collected in the aftermath of both hurricanes (Fig. 4a). We evaluate the accuracy of the composite maps using a 1:1 fit line that represents a perfect match between observed and simulated maximum WLs (Figure 4b). The validation process indicate that Delft3D-FM and 2D HEC-RAS produce maximum WLs in close agreement with available high-water marks in G-Bay, and additionally, both RMSE and MAE agree well with the corresponding metrics reported in similar CF

studies (Huang et al., 2021; Lee et al., 2023; Sebastian et al., 2021; Wing et al., 2019). In addition, we validate peak WLs at selected tide-gauge stations (Figure 3 and Supplementary figures S1 – S2) with respect to CF hazard maps generated in Delft3D-FM for Hurricane Ike and Harvey (Figure 4c and 4d, respectively). Note that we mask out values below 0.10 m in order to enhance visualization of water depths above ground. As expected, the peak WL of Hurricane Ike (~3 m) triggers a relatively large

flood extent and water depth in the lower part of G-Bay whereas the peak WL of Hurricane Harvey (~1.25 m) produces moderate coastal flooding over the same area. However, flood extent and water depth in Harris County are relatively large as compared to those triggered by Hurricane Ike due to the compounding effects of heavy rainfall, extreme river discharge, and storm tides. Such compounding effects are evident at Manchester tide-gage station where freshwater input controls WL dynamics

starting on Aug/27 (Figure 3 h). Also, note that Delft3D-FM constantly outperforms 2D HEC-RAS based on the evaluation metrics and 1:1 line assessment (Figure 3 and 4). Following this, we hereinafter consider Delft3D-FM as the best hydrodynamic model to analyze cascading uncertainty in G-Bay.

### 2.3.4 Model scenarios

    We propose 5 scenarios to analyze the effects of isolated and total uncertainty on CF hazard

assessment for Hurricane Harvey (Table 2). The first scenario focuses on the initial condition of the system including topographic and bathymetric data in coastal DEMs. Recently, Holmquist and Windham-Myers (2022) produced a DEM of relative tidal marsh elevation for the Conterminous United States using land cover classes derived from the 2010 Coastal Change Analysis Program (C-CAP). Since this DEM accounts for elevation errors within coastal wetlands, we conveniently evaluate





uncertainty from the initial condition of the system using the aforementioned DEM and the NOAA's

CUDEM in hydrodynamic simulations (see Section 2.2). The second scenario represents uncertainty

derived from forcing conditions that are often neglected in real-time hurricane-induced flood forecasts

and advisories (CERA, 2023). Specifically, such flood forecasts assume that riverine flow and local

rainfall contributions to flooding are relatively small as compared to that driven by storm surge.

Following this reasoning, we will analyze the impact of local rainfall and riverine flow in CF dynamics

by turning off those two forcing conditions in hydrodynamic simulations.

The third scenario analyzes uncertainty stemming from model parameters including bed

channel and floodplain roughness coefficients associated with land cover classes from the NLCD

(Figure 1a and Table 1). Here, we simulate compound coastal flooding triggered by Hurricane Harvey

using two sets of optimal (calibrated) $n$-values. The first set consists of $n$-values calibrated for

Hurricane Ike (Sep, 2008) whereas the second set comes from the actual model calibration of Hurricane

Harvey (Aug, 2017). The reasoning is that a realistic estimation (or proxy) of calibrated $n$-values for

future flood scenarios consists in those already calibrated for past flood events (Domeneghetti et al.,

2013; Meresa et al., 2021). In this regard, Hurricane Ike is the closest and most relevant CF event that

impacted G-Bay prior to Hurricane Harvey. The fourth scenario represents uncertainty derived from

model structure including model setup and configuration necessary to simulate flood extent and

inundation depth. We therefore analyze this source of uncertainty by comparing model outputs of

Delft3D-FM and 2D HEC-RAS (see Section 2.3). Lastly, the fifth scenario is named "total uncertainty"

as it involves all sources of uncertainty including their cascading effect on CF hazard assessment.




**Figure 4.** Validation of Delft3D-FM and 2D HEC-RAS models in Galveston Bay. (a) Spatial distribution of high-water marks collected in the aftermath of CF events by the U.S. Geological Survey





(USGS). (b) Validation of composite maps with respect to USGS' high-water marks. Score metrics

are calculated for 2D HEC-RAS and Delft3D-FM (in parentheses). CF hazard maps represent

maximum water depths associated with (c) Hurricane Ike and (d) Hurricane Harvey.

**Table 2.** Model scenarios for simulating isolated and total uncertainty in Galveston Bay*.

| | Scenarios | | | | |
|---|---|---|---|---|---|
| **Model description** | **Initial condition (S1)** | **Forcing conditions (S2)** | **Model parameters (S3)** | **Model structure (S4)** | **Total uncertainty (S5)** |
| Hydrodynamic model | Delft3D-FM | Delft3D-FM | Delft3D-FM | *2D HEC-RAS* | *2D HEC-RAS* |
| Calibrated *n*-values | Harvey | Harvey | *Ike* | Harvey | *Ike* |
| Coastal DEM | *CUDEM* | Tidal marsh | Tidal marsh | Tidal marsh | *CUDEM* |
| River flow and rainfall | Turn on | *Turn off* | Turn on | Turn on | *Turn off* |

*Text in *italic* denotes changes with respect to the best model setup for simulating Hurricane Harvey.

**2.4 Regression models**

The proposed model scenarios are further modified to quantify their relative contribution to

total uncertainty using regression models. Here, we report uncertainties in terms of WL residuals

computed between each scenario and the best model setup for simulating Hurricane Harvey. Such

model setup consists in the calibrated Delft3D-FM model that accounts for elevation errors within

coastal wetlands and additionally incorporates the effects or river flow and local rainfall in CF

modeling (Figure 3 and 4). Since 2D HEC-RAS generates raster-based flood maps, we extract WLs at

each grid node of Delft3D-FM to compute the corresponding WL residuals. This in turn helps ensure

consistency in uncertainty analysis over the model domain. Also, note that WL residuals evolve in

time and space and their magnitude attributed to the sources of uncertainty is represented by the four

model scenarios. Therefore, we first compute the maximum WL residual across the model domain for





the entire simulation period (e.g., 10-day window) as well as time-evolving residuals with an interval of 6 hours. This time interval is set to comply with the timing of hurricane advisories of the National Hurricane Center and so enable the construction of 40 datasets within the 10-day simulation window, i.e., WL forecasts and advisories every 6 hours.

### 2.4.1 Multiple-linear regression

WL residuals obtained from physically-based model simulations are used as input features for multiple-linear regression (MLR) and nonlinear models (Figure 5). First, we fit the input features using a MLR model and considered it as a benchmark model to further evaluate the benefits of nonlinear models including physics-informed machine learning (PB-ML). The goal of the MLR model is to estimate total uncertainty as the target variable in terms of WL residuals. Since we are dealing with WL residuals in meters that have a comparable order of magnitude, we do not scale or normalize the input features prior to the fitting process. This is also convenient for evaluation purposes of the fitted model as well as relative importance quantification of each source of uncertainty based on fitted regression weights. We do, however, identify and remove outliers in the input features especially those arising at the edges of the mesh close to the BCs. We use the 'statsmodel API' package in Python to conduct a robust fitting of input features (https://www.statsmodels.org/stable/api.html) and report regression coefficients with the underlying statistical significance (see Section 3.2).



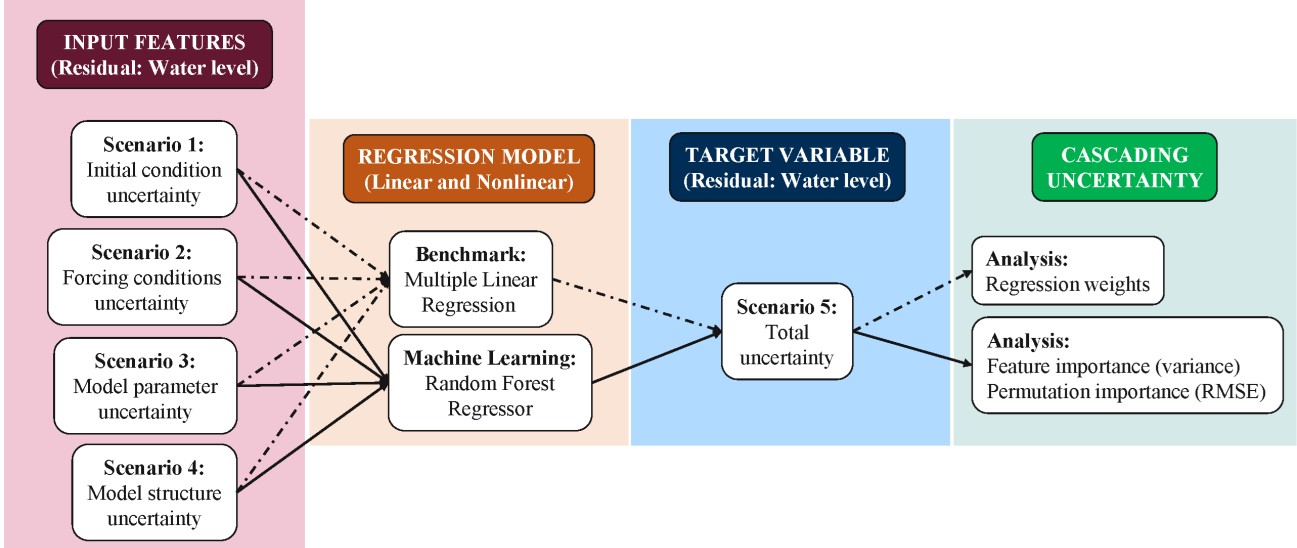

**Figure 5.** Schematic of multiple linear regression and process-based machine learning models to quantify cascading uncertainty. Input features and target variable are reported in terms of water level residuals derived from hydrodynamic simulations of Hurricane Harvey. The target variable contains all sources of uncertainty and their implicit cascading effects.

## 2.4.2 Linked process-based and machine learning models

We conducted a preliminary analysis to identify the best nonlinear ML model that predicts total uncertainty (not shown here for brevity). Among those models, we notice that a random forest regressor model outperforms support vector machine and multiple-layer perceptron models and the latter is also reported in multiple flood studies (Mosavi et al., 2018). Random forest (RF) is a non-parametric ensemble algorithm that builds multiple decision trees based on random bootstrapped samples through replacement (Breiman, 2001). The advantage of RF over other nonlinear regression models relies on its simplicity and easy implementation for efficient regression and classification tasks. Also, RF connects input and target features having complex and nonlinear associations and provides estimates of feature importance to predict the target variable (Alipour et al., 2020; Wang et al., 2015).



We develop a RF regressor model and conduct a thorough model evaluation in Python using the 'scikit-learn tool' package. WL residuals from each scenario are set as input (S1 to S4) and target features (S5) and our analysis is focused on both time-evolving and maximum residuals across the model domain. We split input features into training (80%) and validation datasets (20%) using the total number of data points after outlier removal (e.g., 1'093,501 data points). Those data points represent the number of grid nodes generated in Delft3D-FM (see Section 2.3.1). Next, we conduct hyperparameter tuning to build decision trees and estimate optimal (calibrated) values for each parameter using an HPC system (Table 3).

**Table 3.** Hyperparameter grid and optimal (calibrated) parameter values for the RF regressor model.

| Hyperparameter | Range of values | Total values to be tested | Optimal value |
|---|---|---|---|
| Number of estimators (trees) | 500 – 1000 | 11 | 700 |
| Maximum features | 2 – 4 | 3 | 3 |
| Maximum depth | 10 – 100 | 10 | 60 |
| Minimum sample split | 2 – 10 | 9 | 2 |
| Minimum sample leaf | 2 – 10 | 9 | 2 |
| Bootstrap | True | 1 | True |

We use the scikit-learn tool package to find optimal values and also account for overfitting issues through a cross-validation (CV) process. For this, we generate an initial grid of parameter values within a specified range using the LHS technique (Helton and Davis, 2003). Then, we use a 'k-fold' approach to conduct CV using the training dataset. The number of k-folds is set to 5 and the parameter grid is generated with a fixed seed value to ensure reproducibility. We then use the grid in the 'Randomized Search CV' function to randomly sample a set of hyperparameter values and conduct a



k-fold CV for each combination of values. The number of combinations to be tested is defined by the
'n_iter' parameter that is here set to 100. We repeat the splitting and sampling process for time-
evolving WL residuals and conduct hyperparameter tuning for the resulting 40 datasets using the HPC
system. Nevertheless, we notice that the optimal values associated with each dataset and those reported
in Table 3 lead to similar score metrics in the validation dataset (e.g., $R^2$, R, KGE and RMSE). Also,
the main differences in the score metrics are observed in the third or fourth decimal place (see Section
3.2).

## 3 Results and discussion

### 3.1 Effects of isolated uncertainty

    Scenarios S1 to S4 are designed to analyze the effect of each source of uncertainty on CF
hazard assessment (Figure 6 and Supplementary figure S3). We conveniently display maximum WL
residuals across the model domain where positive or negative values indicate an overestimation or
underestimation of WLs, respectively. Scenario S1 accounts for elevation errors in LiDAR-derived
DEMs that lead to a complex heterogenous pattern with both over- and underestimation of maximum
WL residuals (Figure 6a and 6b). This scenario displays an overestimation of WLs in the tributaries of
the Buffalo Bayou River, the Houston and Galveston navigation channels, and intracoastal waterways
in the lower part of G-Bay. Such an overestimation can be explained by inconsistencies in bathymetric
data between the tidal marsh DEM and the NOAA's CUDEM. In addition, this scenario shows an
underestimation of WLs in the upper part of G-Bay including the head of the Buffalo Bayou's River
tributaries and the San Jacinto River that is surrounded by coastal wetlands (Figure 1a). Wetlands are
natural buffers that dissipate extreme WLs and attenuate storm surge with a rate of 1.7 to 25 cm/km
depending on marsh height, biomass, and storms characteristics (Alizad et al., 2018; Kumbier et al.,
2022; Leonardi et al., 2018). As expected, there is a consistent underestimation of WLs due to vertical



adjustments that lower wetland elevation in the tidal marsh DEM (e.g., with respect to that of CUDEM), reduce wetlands' buffer capacity, and so increase WLs.

Scenario S2 focuses on uncertainty derived from forcing conditions including river flow and local rainfall. This scenario leads to an underestimation of WLs across the model domain (Supplementary figures S3a and S3b). The effect of neglecting forcing conditions on CF hazard assessment is more evident in the northwest side of G-Bay where CF was driven by heavy rainfall and extreme river flow triggering urban flooding in Harris County. In fact, Hurricane Harvey caused urban

flooding in Houston city due to an unprecedent rainfall depth greater than 1.5 m as well as an extreme river flow conveyed by the Buffalo Bayou and San Jacinto River (Blake and Zelinsky, 2018; Sebastian et al., 2021; Valle-Levinson et al., 2020). Scenario S3 analyzes the influence of model parameters that are conventionally calibrated for historical flood events (e.g., Hurricane Ike) and used as a proxy for simulating future flood events (Domeneghetti et al., 2013; Meresa et al., 2021). This scenario exhibits

an overestimation of WLs that is particularly evident in the northwest side of G-Bay (Supplementary figures S3c and S3d). Such an overestimation can be related to the peak WL of Hurricane Ike (~3 m) and Hurricane Harvey (~1.25 m) as well the corresponding calibrated (optimal) $n$-values (Table 1). In general, the $n$-values calibrated for Ike are lower in magnitude than those for Harvey and so they are more effective in reducing bed friction and tidal damping (Bhola et al., 2019; Liu et al., 2018; Mayo

et al., 2014; Pappenberger et al., 2005). Consequently, considering low $n$-values calibrated for Ike to simulate Harvey leads to an overestimation of the storm tide that propagates from the open ocean boundary to the upper part of G-Bay. Note that the peak WL of Ike is 2.4 times larger than that of Harvey and its effect is observed on the maximum WL residuals in the Buffalo Bayou's River tributaries.



**Figure 6.** Effect of isolated uncertainty on compound flood hazard assessment in Galveston Bay. Maximum water level residuals represent model scenarios with uncertainty stemming from (a, b) initial condition and (g, h) model structure. Water level residuals are calculated with respect to the best hydrodynamic model calibrated for Hurricane Harvey. Positive and negative residuals indicate overestimation and underestimation across the model domain, respectively. Right panel shows a zoom-in window over block census groups in Harris County at the northwest side of Galveston Bay.



Scenario S4 accounts for uncertainty derived from model structure and capabilities of 2D HEC-RAS when compared to those of Delft3D-FM. This scenario leads to a complex heterogenous pattern with both over- and underestimation of maximum WL residuals across the model domain (Figure 6c and 6d). Specifically, this scenario displays an overestimation of WLs in the San Jacinto River, Moses and Anahuac lakes, and Freeport whereas an underestimation in Harris County, the Buffalo Bayou River and its tributaries, the Houston and Galveston navigation channels, and intracoastal waterways in the lower part of G-Bay. This complex pattern highlights the capability (or inability) of the hydrodynamic models to account for atmospheric pressure in the conservation of momentum (Eq. 2 and 3) and to capture coastal geomorphological and urban features in the mesh generation process, i.e., structured vs. unstructured grids (Bates, 2023, 2022).

Lastly, scenario S5 is named total uncertainty since it accounts for isolated and cascading effects of the sources of uncertainty on spatiotemporal CF hazard assessment (Supplementary figures S3e and S3f). This scenario displays an overall underestimation of maximum WL residuals in the northwest side of G-Bay which is similar to the pattern observed in scenario S2 (e.g., forcing conditions). Likewise, it displays both over- and underestimation of WL residuals in the lower part of G-Bay that resembles the patterns of scenarios S4 and S1, respectively (e.g., model structure and input data). In contrast, the overestimation pattern of scenario S3 is not visually reflected in scenario S5. This discrepancy is explained in the following section.

## 3.2 Relative contribution of isolated and cascading uncertainty

### 3.2.1 Compound flood hazard assessment

We fit and train MLR and ML models using maximum WL residuals as an indicative of CF hazard in G-Bay (Table 4). Regression weights are statistically significant for all four scenarios ($p$-value $< 0.05$); including S3 in spite of the negative correlation. Also, the Pearson's correlation coefficients reveal a statistically significant linear correlation between total uncertainty and isolated uncertainty (Figure 7). Note that the rank of such correlations, in descending order, agrees well with





the maximum WL residuals and underlying patterns resulting from the model scenarios (Figure 6).
Among them, uncertainty stemming from model parameters has a negative linear correlation with total
uncertainty that partially explains the discrepancy of residual patterns already mentioned in the
previous section. Particularly, a negative correlation suggests that an increase of WL residuals from
scenario S3 results in a decrease of those from scenario S5. Note that the MLR model estimate total
uncertainty with a relatively high accuracy as corroborated by the score metrics, i.e., $R^2 = 0.76$, R =
0.87, RMSE = 0.42 m, and KGE = 0.82 (Figure 7c).


**Table 4.** Multiple-linear regression fitting on maximum water level residuals.

| Scenario (Water level residuals) | Input features | Regression weight | Confidence interval [5%, 95%] |
|---|---|---|---|
| | Intercept | -0.115 | [-0.116, -0.114] |
| S1 | Initial condition | 0.175 | [0.173, 0.176] |
| S2 | Forcing conditions | 1.017 | [1.015, 1.019] |
| S3 | Model parameters | -0.050 | [-0.054, -0.046] |
| S4 | Model structure | 0.681 | [0.679, 0.683] |





**Figure 7.** Isolated and total uncertainty reported in terms of water level residuals [m]. (a, b, d, e) Linear associations with the corresponding Pearson's correlation coefficient. (c, f) Total and predicted





uncertainty obtained from multiple-linear regression and RF regressor models. (g, h, i) Relative

contribution of initial condition, forcing conditions, model parameters, and model structure to total

uncertainty in terms of regression weights, feature and permutation importance.

Overall, the absolute magnitude of regression weights agrees well with the rank derived from

Pearson's correlation coefficients. This suggests that uncertainty stemming from forcing conditions

and model structure are crucial for estimating total uncertainty in CF hazard assessment (Figure 7g).

Other flood studies have shown similar results and demonstrated that uncertainty stemming from

forcing conditions is even more important than that of the remaining sources, especially for flood

prediction and inundation mapping in riverine systems (Alipour et al., 2020; Jafarzadegan et al., 2023;

Pappenberger et al., 2008; Savage et al., 2016). Initial condition of the system and model parameters

are also relevant sources of uncertainty but they show a relatively low regression weight. Note that if

WL residuals of scenarios S1, S3, and S4 are kept invariant, any perturbations of scenario S2 will

result in a nearly identical response of scenario S5 plus a negative offset of 12 cm. Although MLR

models help analyze the influence of each input feature to estimating total uncertainty, they do not

capture 'hidden' associations and/or interactions among the input features. This in turn reduces the

effectiveness of MLR models to characterize cascading effects on total uncertainty.

To overcome this limitation, we determine whether or not the proposed PB-ML model (e.g.,

RF regressor) outperforms the MLR model based on the aforementioned evaluation metrics. In this

regard, the score metrics evince a substantial improvement by the PB-ML model since RMSE

decreases to 0.28 m whereas $R^2$, R, and KGE increase up to 0.90, 0.95, and 0.92, respectively (Figure

7f). Next, we conduct a rank analysis focused on the contribution of each input feature to model's

variance reduction and overall performance for total uncertainty estimation (Figure 7h and 7i,

respectively). In the proposed RF regressor model, feature importance measures the mean decrease of

variance within each selected decision tree whereas permutation importance measures the decrease in

a pre-defined score metric when individual input features are randomly shuffled (Breiman, 2001).

Specifically, the four input features are shuffled multiple times and the RF regressor model is refitted



to estimate their importance in the model's performance. Here, we shuffle the input features 100 times and set RMSE as the evaluation metric to be consistent with the ensemble members and score metrics used in the calibration and validation process (see Section 2.3.2). Notably, the rank of feature importance agrees well with that of the regression weights which in turn indicates that forcing

conditions is not only the main contributor to total uncertainty in CF hazard assessment (56%) but also a key feature for variance reduction (Figure 7h). It is noted that model structure, input conditions, and model parameters contribute to variance reduction to 20%, 18%, and 6%, respectively.

The advantage of permutation over feature importance analysis is that the former method circumvents any overfitting issues by focusing the analysis on validation data (~219k data points).

Also, permutation importance is not biased towards input features with a large number of unique values as compared to feature importance, i.e., high-cardinality. Nevertheless, note that we address potential issues of overfitting and high-cardinality through a 5-fold CV process in the training process (see Section 2.5.2). Similarly, the rank of permutation importance agrees well with both ranks of regression weights and feature importance (Figure 7i). Nevertheless, note that the permutation importance of

forcing conditions (61%) is higher than that of feature importance which can be partially explained by the number of data points used in the computation of the corresponding importance as well as the objective function, i.e., training vs. test datasets and mean variance decrease vs. RMSE. Note that model structure, input, and parameters contribute to the overall models' performance in 24%, 9%, and 6%, respectively. Following these results, we recommend that any efforts for improving CF modeling

and hazard assessment should focus on accounting for all relevant forcing (boundary) conditions and implementing a suitable hydrodynamic model to simulate complex compound coastal flood dynamics. For example, recent studies have shown that estimating water deficits between upstream and downstream flows and distributing such deficits using lateral flows and vertical fluxes (as additional forcing conditions) improve the performance of flood modeling during extreme events (Jafarzadegan

et al., 2021b; Oruc Baci et al., 2023).





### 3.2.2 Compound flood modeling

We track the trajectory of the four sources of uncertainty during the onset, peak, and dissipation of Hurricane Harvey using a 6-hour interval for the entire simulation period (Figure 8a). For this, we fit MLR and train PB-ML models for each of the 40 datasets containing WL residuals and report

relative and cumulative contributions as well as models' performance in terms of $R^2$, R, KGE and RMSE. Model structure and the initial condition of the system are the main sources of total uncertainty at the onset of Hurricane Harvey with a contribution of 60% and 20%, respectively (Figure 8b). However, their relative contribution drops around the peak WL since forcing conditions becomes a more relevant source of uncertainty until the dissipation of WLs, i.e., drastic increase from 10% to

50%. The contribution of model parameters is estimated at 10% and remains almost invariant during the onset, peak, and dissipation of Harvey. Furthermore, the cumulative contribution of each source of uncertainty helps visualize their overall importance for total uncertainty estimation in the simulation period (Figure 8d). In contrast to the rank of contributions established for maximum WL residuals (see Section 3.2.1), note that the model scenario and/or input feature derived from model structure is the

main contributor to variance reduction of the RF regressor model (49%) followed by that of forcing conditions (23%), initial condition (20%), and model parameters (8%).

These results are somehow similar to other studies that analyzed how the influence of forcing (boundary) conditions and model parameters change during flood events (Alipour et al., 2022; Jafarzadegan et al., 2021b; Savage et al., 2016). Although those studies identify forcing conditions as

the most influential factor for flood inundation mapping, uncertainty stemming from model structure is not explicitly analyzed yet recognized as a determinant factor of the results. Lastly, the evaluation metrics computed for the 40 datasets indicate that RF regressor models (dashed line) outperform the benchmark MLR models (solid line) in the simulation period (Figure 8c). Note that $R^2$, R, and KGE display a sudden drop around the peak WL suggesting that MLR models cannot fully characterize total

uncertainty whereas the same evaluation metrics of RF regressor models remain almost invariant due to the ability of nonlinear models to capture any complex interactions as well as cascading effects



arising from the four sources of uncertainty. Likewise, note that RMSE drops to ~11 cm when characterizing cascading uncertainty with RF regressor models. Also, there is a rather constant RMSE of ~25 cm when estimating total uncertainty in terms of WL residuals in the simulation period.


**Figure 8.** Evolution of water level residuals as a proxy of total uncertainty during the onset, peak, and dissipation of Hurricane Harvey. (a) Observed water levels in Morgans Point station located in the middle of G-Bay. (b) Relative contribution of initial condition, forcing conditions, model parameters, and model structure to variance reduction in total uncertainty estimation. (c) Evolution of R$^2$, R, KGE,



and RMSE in the simulation window with solid and dashed lines representing multiple-linear regression and RF regressor models, respectively. (d) Cumulative contribution of the four main sources of uncertainty. The area under de curves represent the average contribution to variance reduction for the entire CF event.

## 4 Conclusions

In the present study, we characterize isolated and cascading uncertainty during the onset, peak, and dissipation of Hurricane Harvey in Galveston Bay, TX. For this, we develop two hydrodynamic models (e.g., Delft3D-FM and 2D HEC-RAS) to simulate compound coastal flooding and conduct compound flood (CF) hazard assessment. The calibrated and validated models help simulate a set of scenarios that reflect uncertainties stemming from initial condition, forcing conditions, model parameters, and model structure. We then train a physics-informed machine learning model (PB-ML) to estimate total uncertainty in terms of water level (WL) residuals and evaluate the model's performance with respect to a benchmark multiple-linear regression (MLR) model. The effects of isolated uncertainty on CF hazard assessment match the spatial patterns observed in the total uncertainty scenario across the model domain; especially for the scenarios that reflect uncertainty from the initial condition of the system, forcing conditions, and model structure. Conversely, the scenarios representing total uncertainty and that from model parameters exhibit a negative correlation resulting in a discrepancy of spatial patterns across the model domain. Nevertheless, we estimate that forcing conditions, model structure, initial condition, and model parameters contribute to variance reduction of the PB-ML model in 56%, 20%, 18%, and 6%, respectively. The latter agrees well with the rank of regression weights estimated with the MLR regression model which help support the conclusion that forcing (boundary) condition is the main contributor to total uncertainty in CF hazard assessment.

Regarding CF modeling, we observe an interplay of relative importance where model structure and the initial condition of the system are the main sources of total uncertainty at the onset of Hurricane

Harvey. Yet, their relative importance drops around the peak WL since forcing conditions becomes a more relevant source of uncertainty until the dissipation of WLs. Also, the importance of model parameters remains almost invariant during the onset, peak, and dissipation of Harvey. Nonetheless, model structure is the main contributor to variance reduction (49%) followed by forcing conditions (23%), initial condition (20%), and model parameters (8%). Lastly, MLR models are not suitable to

characterize total uncertainty since their performance is sensitive to the peak WL as evinced in the evaluation metrics (e.g., RMSE, $R^2$, R, and KGE). Conversely, PB-ML are less sensitive to changes in WL dynamics due to their ability to capture "hidden" interactions and cascading effects arising from the four sources of uncertainty. Following these results, we conclude that PB-ML models are a feasible alternative to conventional statistical methods for characterizing cascading uncertainty in compound

coastal flood modeling and CF hazard assessment. The relative importance of the sources of uncertainty may also vary depending on catchment properties, storm characteristics, and dominant flood drivers. Future work should focus on quantifying cascading and total uncertainty at large-scale and analyzing the effects of the four sources of uncertainty in flood risk assessment (e.g., damage cost).

**Declaration of competing interest**

The authors declare no competing and/or financial interests in this study.

**Acknowledgments**

Partial financial support for this study is provided by the National Science Foundation, CAS-Climate

Program (Award # 480948). Partial support is also provided through a funding awarded to Cooperative Institute for Research to Operations in Hydrology (CIROH) through the NOAA Cooperative Agreement with The University of Alabama (NA22NWS4320003).

**Author contribution**





Muñoz, D.F contributed to conceptualization, methodology, validation, formal analysis, writing the original draft, and visualization. Moftakhari, H. and Moradkhani H. edited the manuscript, and supervised the research project.

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
