# Peer review of "Quantifying cascading uncertainty in compound flood modeling with linked process-based and machine learning models"

_Hydrology and Earth System Sciences, 2024_

## Referee Comment (RC1)

**Reviews for MS #: hess-2024-9 Quantifying cascading uncertainty in compound flood modeling with linked process-based and machine learning models by Muñoz et al.**

This paper deals with uncertainty quantification of compound flooding due to four primary sources of uncertainty (i) Initial condition (ii) forcing uncertainty and lastly (iii) model responses stem from model parameters and structures. For this, a set of hydrodynamic model scenarios were run to quantify the individual and total uncertainty sources. A few places, the analysis requires attention in terms of Methods implemented, otherwise it is written well. Therefore, I suggest major revisions for this manuscript. My comments on the manuscript are as follows:

**A. General Comments**:
1. Line 42: CF events in low-lying areas are typically associated with tropical or extra tropical cyclones for which rainfall-runoff, wind-driven storm surges, total coastal water level including wave set-up and tidal variations, or all of the events concurrently or in a close sequence contribute to the severity compound events (Ganguli & Merz, 2019a; Ganguli & Merz, 2019b).

2. Line 117: multi-model ensemble methods: Kodra et al. (2020) proposed empirical Bayesian model that incorporates skill and consensus based weighing framework to narrow down uncertainty associated with large ensemble of earth system models in the projected climate.

**B. Typing errors**:
3. Line 221: unstructured finite volume grid that consists of triangular 'elements' and not the 'cells'?

4. Line 228: The word, 'in' appeared twice.

5. Figure 6 caption: Effect of individual sources of uncertainty. (a,b) initial condition, (g,h) model structure

6. Line 578: both agrees well with slope of the regression estimate. A linear regression yields two: slope and the intercept terms.

**C. Technical Comments**:
7. Line 245-247: The discharge from the lake upstream and river gauge downstream are estimated simply the sum of two random variables. However, since both random variables are independent, the derived distribution can't be a simple sum – here convolution methods needs to be implemented to quantify sum of two continuous random variables: https://dlsun.github.io/probability/sums-continuous.html

8. Line 91: Chezy's formula that is dependent on surface roughness, Reynold's number of fluid in contact and the mean hydraulic depth.

9. Line 336: 1:1 fit line to be fit of the linear regression

10. Figure 5: In flowchart: Also shows assessment wrt other machine learning methods.

11. Lines 425-430: The comparative assessment with other machine learning methods should also be presented in supplementary.

12. Line 437: How outliers are identified?

13. Sub-section heading 3.1: Effects of Individual and Aggregated Uncertainty

14. Line 513: Why the results of scenario S5 is shown in the Supplementary? It should be presented in the main text. Instead of cascading effects of the sources of uncertainty, the correct term would be total uncertainty considering all four sources that propagate in the system

15. Figure 7: Simply Pearson's $r$ would not be suffice given highly nonlinear relation between individual and total uncertainty, please consider Kendall's tau instead.

16. Table 4: No results shown for scenario S5. How the 95% confidence bounds are obtained in Table 4-please explain in Table footnote.

17. Line 544: Pearson's $r$ doesn't give you rank. Only non-parametric methods are based on rank order transformation. For the former case, $r$ is parameter.

18. One of the crucial steps in uncertainty quantification is narrowing down of uncertainty envelop & the identification of such method that can credibly narrow down the uncertainty. However, no such analyses were presented.

19. Line 613: A PB-ML to outperform ordinary linear MLR is pretty obvious. The assessment wrt other machine learning models should also be discussed.

**References**
Ganguli, P. Merz, B. (2019a). Trends in compound flooding in northwestern Europe during 1901-2014. Geophysical Research Letters. 46(19), 10810-10820.

Ganguli, P. Merz, B. (2019b). Extreme coastal water levels exacerbate fluvial flood hazards in northwestern Europe. Scientific Reports 9(1), 13165

Kodra, E., Bhatia, U., Chatterjee, S., Chen, S., Ganguly, A.R. (2020). Physics-guided probabilistic modelling of extreme precipitation under climate change. Scientific Reports 10(1), 10299.

---

## Referee Comment (RC2)

**Summary**

This study presents an uncertainty analysis for process-based flood models during a compound flood event (Hurricane Harvey). The authors verify different sources of uncertainty and determine which contributes more to the overall error using a machine-learning approach.

**Major Comments**

I would like to congratulate the authors for a wonderful piece of sciene the have in this manuscript. The study is worth of publication given its nature and novelty around compound flood (CF). However, the manuscript needs some improvements to be publishable in this journal. First, the introduction section needs improvements to help the reader follow the manuscript story. First, I would suggest breaking up the model uncertainty sources for process-based models into individual paragraphs instead of grouping them into pairs in two very long paragraphs. Second, I was impressed not to find the study objective/aims within this section. For example, the section ends with a literature review paragraph about previous uncertainty studies instead of the study goals/objectives and the manuscript layout of the following sections. Furthermore, the authors fail to lay out the research gaps this study tries to bridge and thus are unclear why and what they are trying to do. If it was not for the abstract that I read first, I will be completely lost about the study goals based only in the introduction.

Second, the order of some sections could be improved to enhance the story-telling flow of the manuscript. For example, I should know first which hydrodynamic model (Section 2.3) you used instead of the data (Section 2.2) and case study (Section 2.1). This is highlighted in the caption of Figure 1 that mentioned Delft-FM and 2D HEC-RAS but I was unaware that the authors were planning to do two different modeling approaches. This could also be improved if mentioned in the introduction, as suggested above. I recommend talking about the models selected first, then the case study and lastly the data for this section. If not, you should at least create a brief paragraph before 2.1 and give the reader a broader vision of the methods, such as the models to be used, the study area, and any other relevant details, like a summary of this section. Third, the calibration section ( Section 2.3.2) seems quite long and confusing regarding the different calibrating scenarios. I would divide this subsection into three different ones, one for tides only calibration, another for hurricane calibration, and another for hurricanes validation. Also, this could be considered as its section outside of the methods since they include numerical model results. Fourth, a limitation section within the manuscript should be added. Currently, limitations are all over the manuscript, and it would help the reader if they are summarized in a single location.

Lastly, the result and discussion section is very long. I would recommend splitting this into a result and discussion section separately. This will follow the traditional journal articles more, and we can better differentiate the discussion from the results.

**Minor Comments**

- L28: the use of "(56%) 49%" is not clear in the provided context and could confuse the reader. I suggest to revise this statement.
- L32: I will remove the "i.e." and just put it in parenthesis next to the "gross domestic product"., similarly to the below statement of the population percent.
- L35: you should define what is "in the past five years" since it could be from 2023 (probably when you wrote the statement) but the manuscript could be published in 2024.
- L38: I would add "flood" between "coastal drivers" to emphasize the flood hazard. Also, the compound flood definition should be referenced to some of the first publications that studied and defined this in detail, like Bilskie and Hagen (2018).
- L47-65: On this paragraph you mention the three main ways to compute CF. However, you only explain two of them in detail. You should add a couple of sentences describing the hybrid approach since the reader may not be familiar with that term.
- L83-68: while the statement about CERA is truth, the authors should comment that is only of the modes that CERA operates, since it also have a compound flood tools for LA. I will suggest rewriting the statement to highlight the above.
- L90: another source of model uncertainty within the model parameters is the soil moisture (antecedent conditions), and should be briefly discussed in this paragraph, especially if you are talking about compound floods.
- L102: I was expecting that the authors would also include the coupling approach as part of the model structure uncertainty. There is vast literature comparing the different coupling approaches (one-way, two-way, tightly and, fully coupled) for CF and how that affects the results. Regardless, if the authors did not test this option, I would still include it in this paragraph to highlight the potential of an additional uncertainty source.
- Figure 1: what is the purpose of having panel b and c? They look very similar (regarding the topobathy) and there is no discussion about this on the text. Also, the figure caption says that Ike was in 2009 but it should 2008. I would prefer to see the numerical mesh of both models side-by-side than the topobathy.
- L148: the authors should comment why they did selected these two events as case studies. These two events are hurricane and we can classify them like CF events. However, sometimes their impacts does not reflect a CF event. For example, Ike was an event mainly dominated by coastal process flooding, whereas Harvey was the opposite and dominated by the hydrologic process. I would like to see this type of statement in this section.

- L186: it should say "Forcing or boundary conditions", right? Also, WL is already defined, why defined it again?
- L194: I do not see in Figure 1a the HWM from Ike. Is that what the authors are referring to? Please rephase the sentence since if my interpretation was incorrect.
- L191-194: Why do the authors only mention the HWM marks from Ike and not Harvey? I would assume there are multiple reports of flood levels for Harvey that could be used. I would also mention them here.
- L197: why did the authors not use a higher resolution precipitation source, such as the Stage IV dataset from NCEP (https://data.eol.ucar.edu/dataset/21.093), which is at a 4km spatial resolution and available for the US? Please justify your selection since we typically use the ERA5 rainfall data for remote locations that does not have these high resolution datasets. There are even studies in this journal that talk about the inaccurate performance of this dataset (ERA5) for compound flood (https://nhess.copernicus.org/articles/23/3379/2023/). Also, gridded rainfall have proven to be more accurate for flood estimation that rain gauges due to their limited coverage and are mostly use to correct the gridded rainfall products.
- L206-208: I strongly suggest the authors remove the governing equations the models are solving, especially if they are not modifying them directly. They seem to be unnecessary and the authors can reference to other publications that introduce the model and its governing equations.
- L223: you should be consistent with your use regarding G-Bay.
- L232: this statement is repeated, regarding hec-ras capabilities. Same on L256.
- L281-282: The authors should provide the equations used to evaluate the model performance, at least in the appendix or cited from another source that used the same equations.
- L296-297: This seems to be a very vague justification of why they did not run the tidal simulations. There is currently a linux version of HEC-RAS (v 6.1) that can be run in HPC systems and is easy to install. Probably, the justification could be that the Delft can run in parallel while HEC-RAS will run in series within HPC and would take significantly more computer resources and time. Please address this.
- Table 1: Why does the optimal only tides row not have a roughness value for all except open water? It is my understanding that tides were only run in Delft; thus, we should see values here, right?
- Figure 3: I will rearrange the order of the panels in this figure. It would be better for the reader if the panels are group by storm event in each column and by gauge in each row. I would also remove the NOAA tide gauge id and replace it with the location name for easy recognition. The authors have also to shown where in Figure 1a are each of these gauges located since they only have with a start and not the specific gauge name/ID.
- L351-352: rephrase the sentence to mention that Delft is better than HEC-RAS for G-Bay. As it is now, it seems like Delft is the best model in the modeling community for G-Bay.

- Figure 4: move the figure earlier in the text, it should be near line 353. Also, it is not clear the model that was used to create panel c and d, please specify on legend. Why not include both max depth flood maps for each model and hurricane? Why not consider presenting a different plot map based on the flood depths as raster (thus, eliminating the different mesh configurations)?
- L355: why the uncertainty assessment with 5 scenarios are only performed with Harvey? I would expect both so the reader can see if the results are associated with the dominating flood. For example, Ike was mainly driven by surge, while Harvey was hydrologic. Furthermore, why on table 2 you mention that would use Harvey but are using calibrated values from Ike. Similarly, why not consider those 5 scenarios both all of them for HEC-RAS and repeat for all of them Delft, instead than a combination? The text on this section explain well all this, but the table does not, thus potentially confusing the reader.
- Figure 6: those this figure intended to have eight panels? I see reference on the caption to panels (g,h) but only see up to d. I prefer to see all scenarios within the manuscript instead of going back and forth to the supplement figures. I would suggest keeping all the zoom-out maps for all scenarios in the manuscript and moving to supplement the zoom-in maps for all scenarios.

---

## Author Comment (AC1)

**Responses to Reviewers' comments on:**

**"Quantifying cascading uncertainty in compound flood modeling with linked process-based and machine learning models"**

David F. Muñoz, Hamed Moftakhari, and Hamid Moradkhani

**Ms. Ref. No.:** hess-2024-9

**Submitted to:** Hydrology and Earth System Sciences

We really appreciate the thoughtful comments and suggestions from the reviewer to improve the quality of the manuscript. For clarity, we have included the original reviewer's comments in blue text and our point-by-point response in black text.

**Reviewer #1 (RC1)**

A few places, the analysis requires attention in terms of Methods implemented, otherwise it is written well. Therefore, I suggest major revisions for this manuscript.

We thank the reviewer's feedback and further comments to improve this study. Please find below a detailed explanation of how changes are made according to your comments.

A. General Comments:

1. Line 42: CF events in low-lying areas are typically associated with tropical or extra tropical cyclones for which rainfall-runoff, wind-driven storm surges, total coastal water level including wave set-up and tidal variations, or all of the events concurrently or in a close sequence contribute to the severity compound events (Ganguli & Merz, 2019a; Ganguli & Merz, 2019b).

Indeed, we agree that extra-tropical storms such as the Nor'easters in the U.S. Atlantic Coast have the potential to drive CF events. Also, we thank for the important references related to CF in northwestern Europe. Please find below is the revised text.

*"CF events in low-lying areas are typically associated with tropical or extra-tropical cyclones for which rainfall-runoff, wind-driven storm surge, or both can be classified as dominant flood hazard drivers (Bevacqua et al., 2020; Eilander et al., 2020; Ganguli and Merz, 2019a). In addition, the role of waves, tides, and nonlinear interactions on extreme water levels (WLs) can be crucial for the accurate simulation and/or prediction of CF events as reported in several studies (Ganguli and Merz, 2019b; Hsu et al., 2023; Nasr et al., 2021; Serafin et al., 2017)."*

2. Line 117: multi-model ensemble methods: Kodra et al. (2020) proposed empirical Bayesian model that incorporates skill and consensus based weighing framework to narrow down uncertainty associated with large ensemble of earth system models in the projected climate.

Thanks for the sharing the contribution of Kodra et al., 2020 regarding multi-model ensemble methods. We have included this reference in the revised text as indicated below.

*"Those methods include linear associations and first-order second moment approximations (Taylor et al., 2015; Thompson et al., 2008), generalized likelihood estimations (Aronica et al., 2002; Domeneghetti et al., 2013), sensitivity analyses (Alipour et al., 2022; Hall et al., 2005; Savage et al., 2016), multi-model ensemble methods (Duan et al., 2007; Kodra et al., 2020; Madadgar and Moradkhani, 2014; Najafi and Moradkhani, 2016), and data assimilation (Abbaszadeh et al., 2019; Moradkhani et al., 2018; Pathiraja et al., 2018)."*

B. Typing errors:

3. Line 221: unstructured finite volume grid that consists of triangular 'elements' and not the 'cells'?

We respectfully disagree with the reviewer since triangular cells is the correct term when referring to unstructured grids. Below is a schematic of triangular cells extracted from the Delf3D-FM Technical Reference Manual for your consideration.

We now proceed by considering a cell attached to a node $i$ in coordinate frame $(\xi', \eta')$, see Figure 3.6, and define an *optimal* angle $\Phi^{opt}$ between two subsequent edges that are connected to node $i$.

[Figure]

**Figure 3.6:** non-rectangular triangular cell; the dashed cell is an optimal equiangular polygon, while the shaded cell is the resulting cell after scaling in $\eta'$ direction; $\Phi_0$ is the angle of the $\xi'$-axis in the $(\xi, \eta)$-frame

4. Line 228: The word, 'in' appeared twice.

Thanks for catching this. We have revised the text as shown below.

*"Similarly, the second hydrodynamic model is developed in 2D HEC-RAS using an unstructured finite volume grid."*

5. Figure 6 caption: Effect of individual sources of uncertainty. (a,b) initial condition, (g,h) model structure.

Thanks, we have corrected this typo as follows.

*"Figure 6. Effect of isolated uncertainty on compound flood hazard assessment in Galveston Bay. Maximum water level residuals represent model scenarios with uncertainty stemming from (a, b) initial condition and (c, d) model structure. Water level residuals are calculated with respect to the best hydrodynamic model calibrated for Hurricane Harvey. Positive and negative residuals indicate overestimation and underestimation across the model domain, respectively. Right panel shows a zoom-in window over block census groups in Harris County at the northwest side of Galveston Bay."*

5. Line 578: both agrees well with slope of the regression estimate. A linear regression yields two: slope and the intercept terms.

We agree with the reviewer and have clarified this in the text, accordingly.

*"Similarly, the rank of permutation importance agrees well with both the ranks of regression weights (or slope terms) and those derived from feature importance (Figure 7i)."*

C. Technical Comments:

7. Line 245-247: The discharge from the lake upstream and river gauge downstream are estimated simply the sum of two random variables. However, since both random variables are independent, the derived distribution can't be a simple sum – here convolution methods needs to be implemented to quantify sum of two continuous random variables:
https://dlsun.github.io/probability/sumscontinuous.html

We thank the reviewer for raising this important comment. Nevertheless, as explained in the link that you kindly provided, the sum of continuous random variables is necessary for estimating probability density functions (PDFs). This will also help estimate cumulative distribution functions and thereby return design periods. Here, we are simply estimating river discharge for Hurricane Harvey by leveraging available upstream time-series data as discussed in a previous peer-reviewed work in G-Bay (Muñoz et al., 2022). Another feasible alternatively consists in leveraging modeled river discharge from the National Water Model as discussed by Huang et al., (2021). In fact, our method led to comparable results with respect to the cited study in G-Bay. While we acknowledge limitations of our method, such as correctly capturing the timing of the peak flow, the sum of river discharge data is a reasonable proxy for the upstream boundary condition in the San Jacinto River. Moreover, this proxy leads to satisfactory water level simulations as shown in Figure 3 (e to h) and S2.

8. Line 91: Chezy's formula that is dependent on surface roughness, Reynold's number of fluid in contact and the mean hydraulic depth.

We appreciate the reviewer's recommendation. While we do not work directly with the Chezy's formula as per the model configuration in Delft3D-FM (e.g., Manning's equation), we account for the Reynold's number in the eddy viscosity concept for both laminar and turbulent flows, i.e., $v_H$ is the horizontal viscosity parameter in Eq. (2) and (3). The viscosity concept expresses the Reynolds stress component as the product between flow and grid-dependent $v_H$ as well as the corresponding components of the mean rate-of-deformation tensor. We believe this technical

explanation is discussed in detail in any "Open Channel Flow" lectures. Therefore, for the sake of brevity, we limit our analysis to the continuity and momentum equations already included in the manuscript.

**9. Line 336: 1:1 fit line to be fit of the linear regression**

Here, we use a 1:1 line to evaluate the accuracy of simulated water levels using composites of the maximum values. We apologize for the confusion when referring to a 1:1 fit line from a linear regression model. The revise text reads as follows:

*"We evaluate the accuracy of the composite maps by comparing observed and simulated maximum WLs (Figure 4b). Data points that fall along the 1:1 (diagonal) line represent a perfect match between those maximum WLs."*

**10. Figure 5: In flowchart: Also shows assessment wrt other machine learning methods.**

We thank for the suggestion. However, we would like to keep the flowchart as simple as possible by focusing on the random forest (RF) regressor as it outperforms other widely used algorithms including artificial neural networks (ANN) and support vector regressor (SVR).

**11. Lines 425-430: The comparative assessment with other machine learning methods should also be presented in supplementary.**

We believe that such a discussion is unnecessary based on multiple peer-review studies and underlying evidence suggesting that ensemble learning methods (e.g., RF regressor) outperform other machine learning algorithms like SVM and ANN (Mosavi et al., 2018; Chen et al., 2020; Schoppa et al., 2020). For the reviewer's convenience, we include the requested assessment in this response as follows:

We compare the results obtained from the RF regressor with those of ANN (e.g., multilayer perceptron) and SVR (Figure RC1). ANN's and SVR's model parameters are calibrated by following an identical hyperparameter grid approach as described in the RF regressor (Section 2.4.2). The calibrated parameters for ANN are: hidden_layer_sizes=(100), activation='relu', solver='adam', learning_rate='adaptive', learning_rate_init=0.001, and tol=0.001. On the other hand, the calibrated parameters for SVR are: kernel="linear", C=10, gamma="auto", max_iter=500, and tol=0.001. ANN achieves satisfactory results with respect to those of RF regressor in terms of RMSE and both Pearson's and Kendall's correlation coefficients; the latter reported in parentheses. In contrast, SVR did not perform well even with nonlinear kernels such as the radial basis function (RBF).

[Figure]

Figure RC1. Comparison of machine learning models for predicting total uncertainty. Results of RMSE, Pearson's, and Kendall's correlation coefficients are show for ANN (left panel), SVR (middle panel), and RF regressor (right panel). RF regressor outperforms both ANN and SVR.

**12. Line 437: How outliers are identified?**

We agree that an explanation of outlier removal is needed in the manuscript. The revised text reads as follows:

*"In the context of hydrodynamic modeling, outliers are unrealistic WLs emerging around upstream and downstream BC lines as well as the edges of the model domain. Such values are extreme values, either positive or negative, that do not reflect WL dynamics within the model domain. Therefore, we masked out such values using a buffer polygon in ArcGIS and proceed with the training and validation dataset using realistic WLs (e.g., 1'093,501 data points)."*

**13. Sub-section heading 3.1: Effects of Individual and Aggregated Uncertainty**

We thank for the suggestion. The heading has been modified accordingly. We prefer the term isolated and total uncertainty as suggested in your comment # 14.

*"3.1 Effects of isolated and total uncertainty"*

**14. Line 513: Why the results of scenario S5 is shown in the Supplementary? It should be presented in the main text. Instead of cascading effects of the sources of uncertainty, the correct term would be total uncertainty considering all four sources that propagate in the system**

We accept your thoughtful suggestions. Scenario S5 is now included in figure 6 as shown below. The figure caption is also modified to include the term "total uncertainty".

[Figure]

*Figure 6. Effect of isolated and total uncertainty on compound flood hazard assessment in Galveston Bay. Maximum water level residuals represent model scenarios with uncertainty stemming from (a, b) initial condition, (c, d) model structure, and (e, f) total uncertainty. Water level residuals are calculated with respect to the best hydrodynamic model calibrated for Hurricane Harvey. Positive and negative residuals indicate overestimation and underestimation across the model domain, respectively. Right panel shows a zoom-in window over block census groups in Harris County at the northwest side of Galveston Bay.*

15. Figure 7: Simply Pearson's r would not be suffice given highly nonlinear relation between individual and total uncertainty, please consider Kendall's tau instead.

We have included the Kendall's tau correlation coefficient in parentheses to address the reviewer's comment as shown below. Nevertheless, we see that both correlation coefficients lead to an identical conclusion regarding the benefit of machine learning approaches.

[Figure]

*Figure 7. Isolated and total uncertainty reported in terms of water level residuals [m]. (a, b, d, e) Linear associations with the corresponding Pearson's and Kendall's correlation coefficients; the latter in parentheses. (c, f) Total and predicted uncertainty obtained from multiple-linear regression and RF regressor models. (g, h, i) Relative contribution of initial condition, forcing*

*conditions, model parameters, and model structure to total uncertainty in terms of regression weights, feature and permutation importance.*

16. Table 4: No results shown for scenario S5. How the 95% confidence bounds are obtained in Table 4-please explain in Table footnote.

We realize that the reviewer is misinterpreting Table 4. The multiple linear regression model involves scenarios 1 to 4 as independent variables (e.g., sources of uncertainty). Scenario 5 is the dependent variable (e.g., total uncertainty). The 95% confidence intervals are obtained from the 'statsmodel' package available in both Python and R studio. We have included this information in the main text and Table 4, accordingly.

*"We use the 'statsmodel API' package in Python to conduct a robust fitting of input features (https://www.statsmodels.org/stable/api.html) and report regression coefficients with the underlying statistical significance and confidence intervals (see Section 3.2)."*

**Table 4.** *Multiple-linear regression fitting on maximum water level residuals.*

| Scenario (Water level residuals) | Input features | Regression weight | Confidence interval [5%, 95%] |
|---|---|---|---|
| | Intercept | -0.115 | [-0.116, -0.114] |
| S1 | Initial condition | 0.175 | [0.173, 0.176] |
| S2 | Forcing conditions | 1.017 | [1.015, 1.019] |
| S3 | Model parameters | -0.050 | [-0.054, -0.046] |
| S4 | Model structure | 0.681 | [0.679, 0.683] |

*Confidence intervals are obtained from the 'statsmodel' package available in Python.*

17. Line 544: Pearson's r doesn't give you rank. Only non-parametric methods are based on rank order transformation. For the former case, r is parameter.

We apologize for the confusion. We know that Kendall's tau and Spearman's rho are nonparametric methods that measure the association between two variables. In contrast to the Pearson's rho, they both rank data as correctly pointed out by the reviewer. Here, we refer to the rank resulting from the computed coefficients (either Pearson or Kendall) and its agreement with the absolute magnitude of regression weights. We do not refer to the rank of the data themselves. We have clarified the text as follows.

*"Overall, the absolute magnitude of regression weights agrees well with the rank resulting from either Pearson's or Kendall's correlation coefficients."*

18. One of the crucial steps in uncertainty quantification is narrowing down of uncertainty envelop & the identification of such method that can credibly narrow down the uncertainty. However, no such analyses were presented.

We fully agree with the reviewer's comment. However, the main objective of this study is to characterize the sources of uncertainty using process-based and machine learning methods as described in the abstract. Reducing uncertainty using residual learning techniques is the next step (our ongoing work) once the proposed methodology has been validated. To account for the reviewer's comment, we have edited the text in the conclusion section as follows:

*"Following these results, we conclude that PB-ML models are a feasible alternative to conventional statistical methods for characterizing cascading uncertainty in compound coastal flood modeling and CF hazard assessment. The relative importance of the sources of uncertainty may also vary depending on catchment properties, storm characteristics, and dominant flood drivers, i.e., coastal to inland transition zones. Ongoing work is being conducted to effectively reduce uncertainty using residual learning techniques. Also, future work should focus on quantifying and reducing cascading and total uncertainty at large-scale and analyzing the effects of the four sources of uncertainty in flood risk assessment (e.g., damage cost)."*

19. Line 613: A PB-ML to outperform ordinary linear MLR is pretty obvious. The assessment wrt other machine learning models should also be discussed.

As discussed in the comment # 11, we have included the suggested assessment in this response. We believe that a comparison is unnecessary based on multiple peer-review studies and underlying evidence suggesting that RFR outperforms ANN and SVM. Furthermore, our results presented in Figure RC1 support this claim.

**Author's References:**

Chen, W., Li, Y., Xue, W., Shahabi, H., Li, S., Hong, H., et al. (2020). Modeling flood susceptibility using data-driven approaches of naïve Bayes tree, alternating decision tree, and random forest methods. *Science of The Total Environment* 701, 134979. doi: 10.1016/j.scitotenv.2019.134979

Huang, W., Ye, F., Zhang, Y. J., Park, K., Du, J., Moghimi, S., et al. (2021). Compounding factors for extreme flooding around Galveston Bay during Hurricane Harvey. *Ocean Modelling* 158, 101735. doi: 10.1016/j.ocemod.2020.101735

Mosavi, A., Ozturk, P., and Chau, K. (2018). Flood Prediction Using Machine Learning Models: Literature Review. *Water* 10, 1536. doi: 10.3390/w10111536

Muñoz, D. F., Abbaszadeh, P., Moftakhari, H., and Moradkhani, H. (2022). Accounting for uncertainties in compound flood hazard assessment: The value of data assimilation. *Coastal Engineering* 171, 104057. doi: 10.1016/j.coastaleng.2021.104057

Schoppa, L., Disse, M., and Bachmair, S. (2020). Evaluating the performance of random forest for large-scale flood discharge simulation. *Journal of Hydrology* 590, 125531. doi: 10.1016/j.jhydrol.2020.125531

---

## Author Comment (AC2)

**Responses to Reviewers' comments on:**

**"Quantifying cascading uncertainty in compound flood modeling with linked process-based and machine learning models"**

David F. Muñoz, Hamed Moftakhari, and Hamid Moradkhani

**Ms. Ref. No.:** hess-2024-9

**Submitted to:** Hydrology and Earth System Sciences

We really appreciate the thoughtful comments and suggestions from the reviewer to improve the quality of the manuscript. For clarity, we have included the original reviewer's comments in blue text and our point-by-point response in black text.

**Reviewer #2 (RC2)**

I would like to congratulate the authors for a wonderful piece of science they have in this manuscript. The study is worth of publication given its nature and novelty around compound flood (CF). However, the manuscript needs some improvements to be publishable in this journal.

We really appreciate the encouraging words and feedback from the reviewer on the present study. Please find below a detailed explanation of how changes are made according to your comments.

Major Comments:

First, the introduction section needs improvements to help the reader follow the manuscript story. First, I would suggest breaking up the model uncertainty sources for process-based models into individual paragraphs instead of grouping them into pairs in two very long paragraphs.

Thanks for the suggestion. In the revised version, we have described each source of uncertainty using individual paragraphs.

Second, I was impressed not to find the study objective/aims within this section. For example, the section ends with a literature review paragraph about previous uncertainty studies instead of the study goals/objectives and the manuscript layout of the following sections. Furthermore, the authors fail to lay out the research gaps this study tries to bridge and thus are unclear why and what they are trying to do. If it was not for the abstract that I read first, I will be completely lost about the study goals based only in the introduction.

We appreciate the reviewer's thoughtful comments and suggestions. We have now included a paragraph in the end of the introduction section that addresses the reviewer's concern. The text reads as follows:

*"Nevertheless, there is a fundamental gap in terms of understanding the evolution of uncertainty sources in CF modeling as well as their cascading effects propagating in the modeling chain and ultimately leading to total uncertainty. Notably, there is a need of a robust and computationally efficient methodology that enables a proper characterization of spatiotemporal evolution of*

*uncertainty throughout CF events. Here, we aim at characterizing the spatiotemporal evolution of uncertainty in a well-known CF event, namely Hurricane Harvey in Galveston Bay, TX. For this, we develop a PB-ML model framework that combines two different hydrodynamic models as well as (non-) linear regression methods in order to quantify isolated and cascading uncertainty in terms of maximum WL residuals. Also, we leverage the regression models to track the evolution of WL residuals during the onset, peak, and dissipation of Hurricane Harvey. Based on a rigorously trained PB-ML model, we are able to estimate the relative and cumulative contribution of the four sources of uncertainty to total uncertainty over time."*

Second, the order of some sections could be improved to enhance the story-telling flow of the manuscript. For example, I should know first which hydrodynamic model (Section 2.3) you used instead of the data (Section 2.2) and case study (Section 2.1). This is highlighted in the caption of Figure 1 that mentioned Delft-FM and 2D HEC-RAS but I was unaware that the authors were planning to do two different modeling approaches. This could also be improved if mentioned in the introduction, as suggested above. I recommend talking about the models selected first, then the case study and lastly the data for this section. If not, you should at least create a brief paragraph before 2.1 and give the reader a broader vision of the methods, such as the models to be used, the study area, and any other relevant details, like a summary of this section.

Thanks for the suggestion. In the revised manuscript, we have improved the story-telling flow by providing a summary of the hydrodynamic and regression models developed for the study area. The summary is included before section 2.1 as per the reviewer's suggestion and reads as follows:

*"The following sections describe the publicly available data used to develop two different hydrodynamic models for Galveston Bay, namely Delft3D-Flexible Mesh (FM) and the U.S. Army Corps of Engineers' River Analysis System (HEC-RAS), as well as linear and nonlinear regression models. We then introduce the proposed PB-ML framework to characterize uncertainty in CF events, discuss the results, and provide key remarks in the conclusion section."*

Third, the calibration section (Section 2.3.2) seems quite long and confusing regarding the different calibrating scenarios. I would divide this subsection into three different ones, one for tides only calibration, another for hurricane calibration, and another for hurricanes validation. Also, this could be considered as its section outside of the methods since they include numerical model results.

We accept your suggestion to improve the readability of the calibration section. We have now divided section 2.3. into *"2.3.2 Model calibration"* and *"2.3.3 Model validation"*. Although we agree that numerical results could be placed in another section, we first ensure a robust calibration of both Delft3D-FM and HEC-RAS to subsequently characterize cascading and total uncertainty in the following subsections.

Fourth, a limitation section within the manuscript should be added. Currently, limitations are all over the manuscript, and it would help the reader if they are summarized in a single location.

Thanks for the suggestion. As correctly pointed out by the reviewer, we have acknowledged each challenge and limitation throughout the manuscript in order to ensure transparency and

reproducibility of the proposed methodology in other study areas. Moreover, we have justified each decision made by citing relevant and recent literature. Although we see the value of summarizing limitations in another subsection, this would imply rephrasing the text and even extending the length of the manuscript more than it presently is. Also, as per your following comments, we realize that the manuscript is already long and so we decided to elaborate more on other relevant information that you thoughtfully asked us to include.

Lastly, the result and discussion section is very long. I would recommend splitting this into a result and discussion section separately. This will follow the traditional journal articles more, and we can better differentiate the discussion from the results.

Thanks for your recommendation. We did consider splitting Results and Discussion sections as per the reviewer's request. However, we ended up with a much longer Discussion section due to the need of referring to figures and tables and providing the corresponding explanations. From our experience, going back and forth between Results and Discussion sections is not adequate for manuscripts that provide multiple results in the main manuscript and supplementary material. Also, as correctly pointed out by the reviewer, using separate sections follow traditional formats of journal articles that may not be suitable for other studies like this one. We therefore respectfully reject your recommendation.

Minor Comments:

L28: the use of "(56%) 49%" is not clear in the provided context and could confuse the reader. I suggest to revise this statement.

We accept your suggestion. The revised text reads as follows:

*"Model structure and forcing conditions are the main sources of uncertainty in CF modeling and their corresponding model scenarios, or input features, contribute to 56% of variance reduction in the estimation of maximum water level residuals."*

L32: I will remove the "i.e." and just put it in parenthesis next to the "gross domestic product"., similarly to the below statement of the population percent.

We accept your suggestion. The revised text reads as follows:

*"It is estimated that nearly half of the gross domestic product in the U.S. (46% of GPD) is generated in coastal shoreline counties that are frequently exposed to multiple flood hazards (NOAA Digital Coast, 2020)."*

L35: you should define what is "in the past five years" since it could be from 2023 (probably when you wrote the statement) but the manuscript could be published in 2024.

The reference provided in the end of the sentence indirectly suggests the last year considered in the statement. Nevertheless, we have clarified this as follows:

*"In the past five years (2018 – 2023), the National Center for Environmental Information reported 489 fatalities and over $327 billion of total cost damages as a result of tropical cyclones; in which heavy rainfall and storm surge exacerbate coastal flood impacts (NCEI, 2023)."*

L38: I would add "flood" between "coastal drivers" to emphasize the flood hazard. Also, the compound flood definition should be referenced to some of the first publications that studied and defined this in detail, like Bilskie and Hagen (2018).

We accept your suggestions. The revised text reads as follows:

*"Terrestrial and coastal flood drivers of (non-) extreme nature that either coincide or unfold in close succession trigger compound flood (CF) events such as those already evinced in the U.S. history, i.e., Hurricane Katrina (2005), Sandy (2012), Harvey (2017), Florence (2018), Ida (2021), Ian (2022), and Idalia (2023). CF events in low-lying areas are typically associated with tropical or extra-tropical cyclones for which rainfall-runoff, wind-driven storm surge, or both can be classified as dominant flood hazard drivers (Bevacqua et al., 2020; Bilskie and Hagen, 2018; Eilander et al., 2020; Ganguli and Merz, 2019a)."*

L47-65: On this paragraph you mention the three main ways to compute CF. However, you only explain two of them in detail. You should add a couple of sentences describing the hybrid approach since the reader may not be familiar with that term.

Thanks for the suggestion. We have now provided more details of hybrid methods as follows:

*"CF modeling can be performed via multivariate statistical analysis (Bensi et al., 2020; Jalili Pirani and Najafi, 2023; Sadegh et al., 2018), process-based modeling (Bates et al., 2021; Sanders et al., 2023; Santiago-Collazo et al., 2019), and even "hybrid" methods that link statistical and process-based models to alleviate computational burden by focusing on the most likely pair-wise forcing conditions given the statistical dependence among flood drivers (Abbaszadeh et al., 2022; Gori et al., 2020; Moftakhari et al., 2019; Serafin et al., 2019)."*

L83-68: while the statement about CERA is truth, the authors should comment that is only of the modes that CERA operates, since it also have a compound flood tools for LA. I will suggest rewriting the statement to highlight the above.

We accept your suggestion. The revised manuscript acknowledges the pilot study in LA as follows:

*"For example, the Coastal Emergency Risk Assessment (CERA) portal provides real-time storm-surge, wave, and flood guidance for the Gulf and Atlantic Coasts of the U.S. under the assumption that river flow and local rainfall contributions to flooding are relatively small as compared to that driven by storm surge (CERA, 2023). Although this assumption might be valid for non-estuarine regions, ignoring nonlinear interactions among flood drivers in freshwater-influenced stretches of the coast can lead to an underestimation of CF hazards especially in coastal to inland transition zones characterized by tidally-influenced rivers (Bakhtyar et al., 2020; Muñoz et al., 2022b; Yin et al., 2021). Nevertheless, we acknowledge the ongoing work of CERA to incorporating*

*freshwater inflow in CF simulations and flood guidance as demonstrated in a pilot study in Louisiana."*

L90: another source of model uncertainty within the model parameters is the soil moisture (antecedent conditions), and should be briefly discussed in this paragraph, especially if you are talking about compound floods.

We agree with the reviewer in the sense that antecedent soil moisture conditions might influence CF dynamics especially at the onset of flood events. Consequently, soil infiltration capacity becomes an important source of uncertainty stemming from model parameters. However, for practical purposes, modelers assume that soils are completely saturated and even that sewer/drainage systems are working at their full capacity throughout flood episodes. We have included your suggestion in the revised manuscript as follows:

*"Another important source of uncertainty in CF modeling is associated with model parameters such as the antecedent soil moisture condition (e.g., infiltration capacity) and the Manning's roughness coefficient that is present in the bottom stress component of the momentum equation (see Section 2.3). Although soil moisture might influence CF dynamics especially at the onset of flood events, modelers often assume that soils are already saturated for practical purposes. In contrast, Manning's roughness help account for bed friction exerted by vegetation, seabed, riverbed, sinuosity, and irregularity of channel cross-sections (Attari and Hosseini, 2019; Bhola et al., 2019; Yen, 2002). Thus, hydrodynamic models rely on a rigorous static (or dynamic) calibration of roughness coefficients to capture the onset, peak, and dissipation of WLs as well as CF dynamics (Jafarzadegan et al., 2021a; Liu et al., 2018; Mayo et al., 2014)."*

L102: I was expecting that the authors would also include the coupling approach as part of the model structure uncertainty. There is vast literature comparing the different coupling approaches (one-way, two-way, tightly and, fully coupled) for CF and how that affects the results. Regardless, if the authors did not test this option, I would still include it in this paragraph to highlight the potential of an additional uncertainty source.

We completely agree that model coupling falls in the category of model structure uncertainty. As correctly pointed out, we did not test such an option in the present study but rather compared two widely-used hydrodynamic models that can simulate CF events. We acknowledge the aforementioned uncertainty as follows:

*"The fourth source of uncertainty refers to limitations or a priori (theoretical) assumptions that are necessary to simplify the representation of oceanic, hydrological, and meteorological processes in regard to flood generation and routing (Moradkhani et al., 2018; Nearing et al., 2016; Pappenberger et al., 2006). Moreover, uncertainty derived from model structure accounts for model coupling approaches such as one-way, two-way, tightly and fully coupled (Bilskie et al., 2021; Muñoz et al., 2021; Santiago-Collazo et al., 2019) as well as model configuration that refers to inherent "reduced-physics" schemes to solve the conservation of mass and momentum equations (see Section 2.3)."*

Figure 1: what is the purpose of having panel b and c? They look very similar (regarding the topobathy) and there is no discussion about this on the text. Also, the figure caption says that Ike was in 2009 but it should 2008. I would prefer to see the numerical mesh of both models side-by-side than the topobathy.

Thanks for catching the error in the year of Hurricane Ike. The main purpose of panel b and c is to show the reader that the underlying mesh of both models is adequate to capture morphological and hydrodynamic features through spatial interpolation. Moreover, it enables an adequate simulation of CF dynamics as shown in figures 3 and 4 and figures S1, S2, and S3 in the supplementary material. The mesh size varies from 3 km in the open ocean to 5 m in Harris County. Therefore, displaying the mesh (even with colored edges) makes the figure illegible. For that reason, we decided to display the interpolated topobathy. We have clarified the reviewer's concern in the figure caption as follows:

[Figure]

*Figure 1. Model domain of Galveston Bay, TX. (a) Tide-gauge stations and land cover categories derived from the National Land Cover Database. Solid and dashed lines illustrate the best tracks of Hurricane Ike (Sep, 20089) and Harvey (Aug, 2017), respectively. Topography and bathymetry of the study area interpolated in (b) Delft3D-FM and (c) 2D HEC-RAS models are almost identical suggesting that the underlying mesh can capture key morphological and hydrodynamic features.*

L148: the authors should comment why they did select these two events as case studies. These two events are hurricane and we can classify them like CF events. However, sometimes their impacts do not reflect a CF event. For example, Ike was an event mainly dominated by coastal process

flooding, whereas Harvey was the opposite and dominated by the hydrologic process. I would like to see this type of statement in this section.

Thanks for your suggestion. As correctly pointed out by the reviewer, tropical and extratropical cyclones are the main physical drivers of compound flooding. Often times, either terrestrial or coastal flood drivers dominate the compound flood process at the global scale (Eilander et al., 2020). We selected those hurricanes for two main reasons as explained in the revised manuscript.

*"We simulate two CF events in G-Bay, namely Hurricane Ike and Hurricane Harvey, that hit the Gulf of Mexico in September 2008 and August 2017, respectively (Figure 1a). Those hurricanes are selected because they were not only the most recent and relevant CF events in G-Bay but also driven by dominant coastal (storm surge) and terrestrial (rainfall-runoff) flood drivers, respectively".*

L186: it should say "Forcing or boundary conditions", right? Also, WL is already defined, why defined it again?

Thanks for catching this typo. Below is the revised sentence.

*"Forcing or boundary conditions (BCs) consist of time-series data of WL and river discharge that are obtained from the NOAA's Tide & Currents portal (https://tidesandcurrents.noaa.gov/) and the USGS' National Water Dashboard (https://dashboard.waterdata.usgs.gov/), respectively."*

L194: I do not see in Figure 1a the HWM from Ike. Is that what the authors are referring to? Please rephase the sentence since if my interpretation was incorrect.

We apologize for the confusion. High-water marks for both Hurricanes are shown in Figure 4a. We have now rephrased the sentence as follows:

*"To evaluate hydrodynamic model's performance, we leverage survey data from a temporary monitoring network deployed for Hurricane Ike, i.e., water pressure sensors (East et al., 2008), and post-flood high-water marks from the USGS's Flood Event Viewer (https://stn.wim.usgs.gov/fev/) for Hurricane Harvey (Figure 4a)."*

L191-194: Why do the authors only mention the HWM marks from Ike and not Harvey? I would assume there are multiple reports of flood levels for Harvey that could be used. I would also mention them here.

We considered available high-water marks for Hurricane Ike and Harvey. This was clarified in the previous comment. Thanks for catching this.

L197: why did the authors not use a higher resolution precipitation source, such as the Stage IV dataset from NCEP (https://data.eol.ucar.edu/dataset/21.093), which is at a 4km spatial resolution and available for the US? Please justify your selection since we typically use the ERA5 rainfall data for remote locations that does not have these high resolution datasets. There are even studies in this journal that talk about the inaccurate performance of this dataset (ERA5) for compound flood (https://nhess.copernicus.org/articles/23/3379/2023/). Also, gridded rainfall have proven to

be more accurate for flood estimation that rain gauges due to their limited coverage and are mostly use to correct the gridded rainfall products.

We thank the reviewer for raising this important comment. While we are aware of higher resolution precipitation data from NCEP, Harris County Flood Control District (HCFCD, https://www.harriscountyfws.org/) does possess a very dense rain gauge network (< 4 km resolution). Therefore, we conduct a spatial interpolation to construct gridded precipitation data over time with a spatial resolution of 1 km (Figure RC2). This was also proven to be effective in another peer-review study in G-Bay (Sebastian et al., 2021). ERA-5 rainfall data helps complement HCFCD interpolated data outside the Harris County where the mesh is coarser. That way, any potential errors in CF modeling derived from the coarse ERA-5 data are not reflected in the area of interest.

[Figure]

Figure RC2. Rain gauge network of Harris County.

L206-208: I strongly suggest the authors remove the governing equations the models are solving, especially if they are not modifying them directly. They seem to be unnecessary and the authors can reference to other publications that introduce the model and its governing equations.

We respectfully disagree with the reviewer's point of view. Throughout the manuscript, we refer to the governing equations for supporting our discussions and providing explanations of model structure errors as well as complex CF patterns shown in Figure 6. Also, we consider important to present those equations given that we explicitly work with two hydrodynamic models and underlying uncertainty.

L223: you should be consistent with your use regarding G-Bay.

Thanks for catching this. We have revised this error as follows:

*"Those features include the G-Bay entrance, artificial channels in Houston, intracoastal waterways, lateral floodplains, wetland regions, and bottleneck-like connections between G-Bay and both the Buffalo Bayou River and San Jacinto River (Figure 1c)."*

L232: this statement is repeated, regarding hec-ras capabilities. Same on L256.

We believe this is important to emphasize given our focus on uncertainties stemming from model structure.

L281-282: The authors should provide the equations used to evaluate the model performance, at least in the appendix or cited from another source that used the same equations.

We accept your suggestion. These equations have been provided in another study in G-Bay. The revised text reads as follows:

*"Then, we identify an optimal (calibrated) value for the "open water" category that achieves the lowest Root-Mean Square Error (RMSE) and Mean Absolute Error (MAE) as well as the highest Kling-Gupta Efficiency (KGE) and Nash-Sutcliffe Efficiency (NSE). These metrics and underlying equations have been presented in another study in G-Bay (Muñoz et al., 2022a)."*

L296-297: This seems to be a very vague justification of why they did not run the tidal simulations. There is currently a linux version of HEC-RAS (v 6.1) that can be run in HPC systems and is easy to install. Probably, the justification could be that the Delft can run in parallel while HEC-RAS will run in series within HPC and would take significantly more computer resources and time. Please address this.

Thanks for your suggestion. We have addressed your comment in the revised manuscript as follows:

*"Note that we only evaluate tidal propagation using Delft3D-FM model as this software is installed in our HPC system whereas 2D HEC-RAS is run in a desktop computer. Moreover, the former model can run in parallel while recent 2D HEC-RAS versions in Linux run in series within the HPC system. This in turn would take significantly more computational resources and time to accomplish 100 ensemble model realizations for a 1-year simulation window."*

Table 1: Why does the optimal only tides row not have a roughness value for all except open water? It is my understanding that tides were only run in Delft; thus, we should see values here, right?

The reviewer is correct. We ran Delft3D-FM for tide-only with the goal of finding the optimal Manning's roughness (n) value for open water (Figure 1a). The remaining n-values corresponding to the other land cover classes were not calibrated and rather set to the same n-value (0.015). We did not populate Table 1 with "n = 0.015" as to avoid confusion regarding the optimal values for Hurricane Ike and Harvey.

Figure 3: I will rearrange the order of the panels in this figure. It would be better for the reader if the panels are group by storm event in each column and by gauge in each row. I would also remove

the NOAA tide gauge id and replace it with the location name for easy recognition. The authors have also to shown where in Figure 1a are each of these gauges located since they only have with a start and not the specific gauge name/ID.

We appreciate the reviewer's suggestion. In the revised manuscript, we have included the location names in Figure 1 and 3 as per your request. We believe that the current order of the panels is adequate since we are not comparing water levels per tide-gauge station. Also, rearranging the panels by storm event in each column would imply adjusting the range of the y-axis in order to allow for comparison purposes.

[Figure]

*Figure 1. Model domain of Galveston Bay, TX. (a) Tide-gauge stations and land cover categories derived from the National Land Cover Database. Solid and dashed lines illustrate the best tracks of Hurricane Ike (Sep, 2009) and Harvey (Aug, 2017), respectively. Topography and bathymetry of the study area interpolated in (b) Delft3D-FM and (c) 2D HEC-RAS models are almost identical suggesting that the underlying mesh can capture key morphological and hydrodynamic features.*

[Figure]

*Figure 3. Model calibration at selected tide-gauge stations in Galveston Bay. Model performance is evaluated in terms of RMSE, NSE, and KGE for (a-d) Hurricane Ike, and (e-h) Hurricane Harvey. Color code indicate score metrics for Delft3D-FM (black) and 2D HEC-RAS (blue).*

L351-352: rephrase the sentence to mention that Delft is better than HEC-RAS for GBay. As it is now, it seems like Delft is the best model in the modeling community for G-Bay.

We accept your suggestion. The revised text reads as follows:

*"Following this, we hereinafter consider Delft3D-FM as the best hydrodynamic model to analyze cascading uncertainty in G-Bay with respect to 2D HEC-RAS."*

Figure 4: move the figure earlier in the text, it should be near line 353.

As the reviewer might realize, the position of figures and tables is ultimately handled by the editorial team. Nevertheless, in the revised manuscript, the position of Figure 4 is conveniently set to cover an entire page in order to preserve the resolution of subpanels to 300 dpi.

Also, it is not clear the model that was used to create panel c and d, please specify on legend.

We accept your suggestion and have now modified the figure caption as follows.

*"Figure 4. Validation of Delft3D-FM and 2D HEC-RAS models in Galveston Bay. (a) Spatial distribution of high-water marks collected in the aftermath of CF events by the U.S. Geological Survey (USGS). (b) Validation of composite maps with respect to USGS' high-water marks. Score metrics are calculated for 2D HEC-RAS and Delft3D-FM (in parentheses). CF hazard maps represent maximum water depths computed with Delft3D-FM and corresponding to (c) Hurricane Ike and (d) Hurricane Harvey."*

Why not include both max depth flood maps for each model and hurricane? Why not consider presenting a different plot map based on the flood depths as raster (thus, eliminating the different mesh configurations)?

Since we have demonstrated that Delft3D-FM outperforms 2D HEC-RAS, there is no point of generating additional flood maps based on the latter model. Also, we argue that the use of either vector or raster formats for displaying flood depths is irrelevant and will not influence the general results reported in this study.

L355: why the uncertainty assessment with 5 scenarios is only performed with Harvey? I would expect both so the reader can see if the results are associated with the dominating flood. For example, Ike was mainly driven by surge, while Harvey was hydrologic. Furthermore, why on table 2 you mention that would use Harvey but are using calibrated values from Ike. Similarly, why not consider those 5 scenarios both all of them for HEC-RAS and repeat for all of them Delft, instead than a combination? The text on this section explains well all this, but the table does not, thus potentially confusing the reader.

The assessment performed with Hurricane Harvey is to demonstrate that model parameters calibrated for past CF events like Hurricane Ike and used to predict "future" events (e.g., Hurricane Harvey) introduce uncertainty that propagates in the modeling chain. Hurricanes Ike and Harvey are key examples of how subsequent CF events can be dominated by different flood drivers; hence, pre-calibrated model parameters are not always suitable for prediction purposes even though they are considered the "best" available option. Table 2 reflects the aforementioned explanation. Another source of uncertainty stems from model structure. Therefore, we first proved that Delft3D-FM outperforms 2D HEC-RAS and then we selected the latter model to propagate such an uncertainty. Although we could have analyzed uncertainty from model structure using individual hydrodynamic models (see Muñoz et al., (2022), Jafarzadegan et al., (2021), among others), we aim at considering different model configurations and capabilities when assessing cascading and total uncertainty. As correctly mentioned by the reviewer, the original text already explains this in detail and so we believe that Table 2 may not lead to any confusion for the reader.

Figure 6: those this figure intended to have eight panels? I see reference on the caption to panels (g,h) but only see up to d. I prefer to see all scenarios within the manuscript instead of going back and forth to the supplement figures. I would suggest keeping all the zoom-out maps for all scenarios in the manuscript and moving to supplement the zoom-in maps for all scenarios.

Thanks for catching this error. As per request of reviewer 1 (RC1), we have modified Figure 6 as follows. Also, with all due respect, your suggestion is rather contradictory since having similar maps with and without zoom-in sections would also imply going back and forth to the supplementary material.

[Figure]

*Figure 6. Effect of isolated and total uncertainty on compound flood hazard assessment in Galveston Bay. Maximum water level residuals represent model scenarios with uncertainty*

*stemming from (a, b) initial condition, (c, d) model structure, and (e, f) total uncertainty. Water level residuals are calculated with respect to the best hydrodynamic model calibrated for Hurricane Harvey. Positive and negative residuals indicate overestimation and underestimation across the model domain, respectively. Right panel shows a zoom-in window over block census groups in Harris County at the northwest side of Galveston Bay.*

**Author's References:**

Eilander, D., Couasnon, A., Ikeuchi, H., Muis, S., Yamazaki, D., Winsemius, H., et al. (2020). The effect of surge on riverine flood hazard and impact in deltas globally. *Environ. Res. Lett.* doi: 10.1088/1748-9326/ab8ca6

Jafarzadegan, K., Abbaszadeh, P., and Moradkhani, H. (2021). Sequential data assimilation for real-time probabilistic flood inundation mapping. *Hydrology and Earth System Sciences* 25, 4995–5011. doi: 10.5194/hess-25-4995-2021

Muñoz, D. F., Abbaszadeh, P., Moftakhari, H., and Moradkhani, H. (2022). Accounting for uncertainties in compound flood hazard assessment: The value of data assimilation. *Coastal Engineering* 171, 104057. doi: 10.1016/j.coastaleng.2021.104057

Sebastian, A., Bader, D. J., Nederhoff, C. M., Leijnse, T. W. B., Bricker, J. D., and Aarninkhof, S. G. J. (2021). Hindcast of pluvial, fluvial, and coastal flood damage in Houston, Texas during Hurricane Harvey (2017) using SFINCS. *Nat Hazards*. doi: 10.1007/s11069-021-04922-3

---

## Author Response (AR2)

**Responses to Editor's and Reviewers' comments on:**

**"Quantifying cascading uncertainty in compound flood modeling with linked process-based and machine learning models"**

David F. Muñoz, Hamed Moftakhari, and Hamid Moradkhani

**Ms. Ref. No.:** hess-2024-9

**Submitted to:** Hydrology and Earth System Sciences

**Editor (EC)**

Thank you for re-submitting your manuscript "Quantifying cascading uncertainty in compound flood modeling with linked process-based and machine learning models".

One reviewer has provided reviews and comments on the new version of the manuscript. The reviewer praised the outstanding quality achieved in your work, both in terms of its scientific rigor and its presentation. Therefore, the manuscript can now be accepted for publication in Hydrology and Earth System Sciences (HESS) journal, before addressing the following two minor technical suggestions:

- The summary of the hydrodynamic and regression models developed for the study area was not in the revised version before section 2.1, as the authors say in the response document.
- Add to the manuscript text the authors' explanation/justification of the rainfall input source (Harris County Rain Gauge +ERA 5) and not just on the response to reviewers since the readers will ask this same question.

Therefore, I would like to invite you to submit a revised version of your manuscript, addressing these indications, as suggested in the reviewer's report. I look forward to seeing the next version of your manuscript which I will not send out for further review.

Thanks for the opportunity to submit the final revised version of the manuscript. We have now addressed the two minor technical suggestions suggested by the reviewer.

On behalf of the co-authors,

David F. Muñoz, Ph.D.

Assistant Professor at Virginia Tech